# Empowering EFL teachers' perceptions of generative AI-mediated self-professionalism

Mohd Nazim [1], Ali Abbas Falah Alzubi[2]*

1 English Department, College of Languages and Translation, Najran University, Najran, Saudi Arabia,
2 Department of English, College of Languages and Translation, Najran University, Najran, Saudi Arabia

* aliyarmouk2004@gmail.com

**Editor:** Ömer Gökhan Ulum, Mersin University: Mersin Universitesi, TÜRKIYE

## Abstract

This study intends to empower English as a Foreign Language (EFL) teachers' perceptions of generative artificial intelligence (AI)-mediated self-professionalism in engagement, attitudes, constraints, and solutions. Employing the mixed methods research design, the researchers collected data from male and female teachers (N=278) of eight public universities, utilizing convenience sampling and a set of instruments: a questionnaire and a semi-structured interview. The data analysis combined quantitative and qualitative methods, using SPSS version 26 for statistical analysis (Pearson correlation, Cronbach's alpha, means, standard deviations), and thematic analysis for qualitative data, with data triangulation employed to compare questionnaire and interview responses for a comprehensive understanding of EFL teachers' engagement with generative AI. The results revealed that the study sample engaged in self-professionalism at a medium level, yet they hold high attitudes toward generative AI-mediated self-professionalism. In addition, the content analysis exhibited several constraints, including technological competence and AI literacy, AI-generated content reliability and accuracy, ethical issues, and encroachment on professional autonomy. Moreover, the respondents proposed solutions such as offering AI-driven training programs, establishing clear ethical guidelines and protocols, emphasizing AI as a supplementary tool rather than a substitute, and implementing impartial access mechanisms for AI content to strengthen EFL teachers' self-professionalism mediated by generative AI. Studies in the context of generative AI-driven self-professionalism appear limited, particularly in the context of Arab higher education institutions. This dearth of research presents an opportunity for the current study to make significant improvements in contributing innovative insights to the EFL teachers' self-professionalism landscape.

## Introduction

Self-professionalism, in the present study context, or teacher professionalism, a cornerstone of pedagogy, extends beyond just excelling in delivering course

**Data availability statement:** The data underlying the results presented in the study are available from 10.6084/m9.figshare.27629505.

**Funding:** This research project was financed by the Deanship of Graduate Studies and Scientific Research at Najran University, Saudi Arabia through a grant (NU/GP/SEHRC/13/404-7). The funder had no role in the study design, data collection and analysis, the decision to publish, or the preparation of the manuscript.

content [1,2]. It implies the standard of excellence, proficiency, and commitment that an instructor should possess when performing duties [3,4]. Self-professionalism, also known as teacher professionalism, encompasses effective instruction, lesson planning, understanding student needs, maintaining classroom order, and upholding ethical standards [5]. All these aspects are significant to a teacher's professional development as they contribute to all academic stakeholders' intellectual, social, and emotional development [6]. Teacher professionalism, according to [7, 8, 9], is the educators' competence and dedication to impart high-quality, long-lasting knowledge to the next generation through consistently working to achieve the pertinent educational standards and be abreast of the most recent advancements in education.

Traditionally, teachers, including EFL practitioners, engaged in various activities such as workshops, conferences, and peer coaching to enhance their pedagogical skills [10]. However, these traditional approaches have faced criticism, with scholars arguing that they are passive and offer teachers limited benefits. Diaz-Maggioli [11] and Ahmad and Shah [12] suggest that, traditionally, teacher education lacks sensitivity to context or individual needs, as instructors are seen primarily as recipients of knowledge from subject matter experts.

In contrast, contemporary training and development, also known as professional or self-development, aims to support teachers in fostering professionalism and advancing their knowledge and skills. Richards and Farrell [13] use *training* and *development* interchangeably, suggesting that training involves preparing for the teaching task, adapting to teaching environments, modifying content, and grouping learners. Development, on the other hand, requires teachers to reflect on their practice, understand their identity, and consider the contexts in which they teach. Professional development enhances teachers' productivity, deepens their understanding, and transforms the academic environment. Assia [14] suggests that professional development enables teachers to boost their productivity and knowledge, leading to positive changes in education systems. Guskey [15] views professional development as a deliberate effort to change classroom practices, attitudes, perceptions, and student outcomes. Schlager and Fusco [16] emphasize that professional development should be ongoing and context-sensitive, tailored to individual needs and situations. Similarly, Avalos [17] highlights the importance of professional development in fostering teachers' progress and empowerment. Bailey et al. [18] note that teaching is often seen as an individual endeavor, and teacher development is about self-directed growth. Teachers set their objectives, choose their paths, monitor their progress, and draw their conclusions, leading to a sense of self-professionalism. These attributes, in the context of this study, contribute to the development of teachers' professional identity and practice.

EFL teachers must enhance their self-professionalism by promoting context-sensitive growth, reflective practice, and transformative pedagogical approaches [19, 20, 21]. The shift from traditional teacher training activities to ongoing professional development initiatives is further complemented by the recent emergence of generative AI tools like ChatGPT. ChatGPT is a software application that uses AI to

create text and can function as a virtual assistant or chatbot, communicating with people in multiple languages and on various subjects [22]. Generative AI tools like ChatGPT are redefining professional practices, presenting unique opportunities and challenges for educators [23]. Teachers are required to enhance their instructional skills and foster and evolve their capacity for self-reflection and inspiration. Scholars agree that AI has revolutionized the academic spectrum through its innovative and transformative technologies, empowering teachers to contribute more greatly than conventional teaching and learning strategies [24].

A review of recent studies reveals a growing interest in how AI technologies, including generative AI, impact teacher professionalism and professional development. With the rapid integration of technology into education, Cukurova et al. [25] argue that the rise of generative AI and its applications, such as ChatGPT, presents a crucial opportunity for educators to reconsider their pedagogical approaches and commit to ongoing professional development. Additionally, Chen et al. [26] observe that AI can help teachers improve their instructional efficiency and enhance pedagogical standards. Moreover, Yajuan et al. [27] suggest that foreign language instructors should not become overconfident or underestimate the impact of AI on traditional education. They should objectively assess AI's benefits, analyze its limitations, adapt to opportunities and challenges, engage in active teaching practices, and embrace innovation in teaching concepts and methods. Therefore, EFL teachers must empower themselves to advance and innovate their pedagogy to reinforce practical teaching and learning practices within and beyond classroom contexts. Omar [28] argues that teachers must engage with educational innovation both inside and outside the academic environment through self-professionalism and cutting-edge technological and AI-driven professional development initiatives. These engagements are fundamental to every effective educational system as they focus on teachers' professionalism in classroom instruction and keeping up with ongoing pedagogical projects in the academic world.

Despite the rise in AI-assisted professional development, there is a distinct lack of research on generative AI's role in EFL teacher professionalism, especially within Arabic-speaking regions. This study addresses this gap by considering the Saudi Vision 2030 and the ongoing transformation of education within Saudi Arabia. Al-Shehria and Gharamah [29] note that "the Kingdom's Vision (2030) included a development plan that focuses on an integrated package of programs to develop the educational environment" (p. 149). They also highlight that "it also focuses on developing teaching methods and raising the capacities of teachers" (p. 149).

The researchers aim to achieve the study's objective through a comprehensive analysis of EFL teachers' perceptions regarding the integration of generative AI in their professional development. This study explores the association between EFL teachers' self-professionalism and generative AI technologies. Hence, this study seeks to enhance EFL teachers' perceptions of generative AI-mediated self-professionalism, focusing on engagement, attitudes, constraints, and solutions. By exploring engagement, attitudes, constraints, and potential solutions, it looks at how AI might enhance teachers' professional development. The study is distinctive because it aligns with Saudi Vision 2030, a national framework that prioritizes educational reforms and the enhancement of teaching methodologies. The focus on generative AI in EFL teacher professionalism, the contextual relevance to Saudi Vision 2030, the in-depth examination of teachers' experiences, the proactive approach to issues and solutions, the emphasis on reflective practice, and the potential impact on educational policies are some of the research's distinctive features. Moreover, the current study offers valuable insights for EFL teachers on how to effectively integrate AI into their instruction. By better understanding the constraints and solutions of integrating AI into classrooms, EFL teachers and education technology researchers may be able to utilize professional development initiatives more effectively. Additionally, the study promotes passive models to innovative, context-sensitive strategies by highlighting introspective and AI-mediated self-professionalism. This study not only aims to explore the role of AI in EFL teacher professionalism but also to influence educational practices in line with national reform initiatives. The results aspire to have an impact on Saudi Arabia's and other Arabic-speaking countries' educational policies and practices. The study adds to the discussion over instructional strategies and the direction of education in a digital age. Hopefully, stakeholders, including EFL teachers, educators, students, and researchers investigating educational technology, particularly

generative AI-mediated self-professionalism, will find significance in the findings of this study. Henceforth, the study aims to achieve the following objectives:

1. assessing the extent of generative AI-mediated self-professionalism activities that EFL teachers engage in.

2. identifying EFL teachers' attitudes toward generative AI-mediated self-professionalism engagement.

3. finding out the constraints the faculty members face when they engage in generative AI-mediated self-professionalism activities.

4. determining the solutions to make the generative AI-mediated self-professionalism engagement effective.

## Review of the literature

### Teacher's professional development

Numerous studies have explored various aspects of teachers' self-professionalism or professional development, investigating a wide range of variables, circumstances, factors, and contexts. For instance, Mahdi [30] delved into the impact of continuous professional development (CPD) on both institutional and personal levels, examining teachers' perceptions of CPD effectiveness. The findings underscored the pivotal role of professional development in shaping instructional practices, highlighting a disconnect between individual needs and CPD policies, such as a lack of ICT training.

Similarly, Hervie and Winful [31] explored how teacher development and training could enhance instructional effectiveness. Their research revealed that inadequate in-service training, a dearth of instructional resources, insufficient incentives and motivation, and inadequate supervision contributed to suboptimal performance among instructors. Recommendations included enhancing the Ghana Education Service's in-service training and development policy, conducting regular assessments of learning needs before designing teacher training programs, and allocating funding for more frequent in-service teacher training sessions. Additionally, boosting teachers' morale was deemed essential for motivating them to perform at their best. In a qualitative study, Qadhi and Floyd [32] investigated the perspectives and experiences of continuing professional development (CPD). They found that while all participants acknowledged the value of CPD for lifelong learning and professional development, their views on CPD varied.

Moreover, Badia [33] also noted that teachers generally agreed that teacher assessment and appraisal procedures could promote professional growth and teaching quality. However, they expressed dissatisfaction with existing evaluation plans, citing a lack of objectivity in the appraisal system and calling for a fair evaluation process. Furthermore, Ouardani [34] highlighted concerns about the quality of continuing professional development (CPD) activities for EFL teachers. Despite the growing interest in CPD and its role in improving learning outcomes, program designers faced significant challenges in delivering high-quality professional development opportunities for teachers.

### AI-mediated teacher's self-professionalism

The ever-evolving landscape of academia undergoes constant transformation, with the emergence of generative AI standing out as a crucial element, especially in enhancing teachers' self-professionalism [35,36,37]. Therefore, it is crucial to explore insights from available studies, each offering distinct perspectives on the role that AI plays in promoting the professional growth of teachers.

The existing literature underscores the potential for AI to improve teaching and learning outcomes in higher education, particularly in the EFL context [38, 39, 40, 41, 42, 43, 44, 45]. However, it indicates persistent challenges concerning training, integration, and awareness. For example, Al-Dosari [46] examined the use of AI in faculty members' professional development, focusing on its use, potential for development, implementation procedures, and acceptance constraints. Results indicated a lack of skill development but a moderate integration of AI for innovation and excellence. Improvements in teaching and learning outcomes were significantly correlated with increased utilization of AI. Participants acknowledged

AI's ability to provide adaptive instruction in line with changing curricula and increase student engagement. However, insufficient training in analytical skills impaired inadequacies in institutional vision. Despite difficulties, the use of AI raised the standard of education, with the main barrier being the lack of training materials.

Additionally, Razia et al. [47] investigated the relationship between AI and its uses in higher education. The study identified top Arab universities and used an online survey to assess AI applications. Important conclusions emphasized the value of knowledge management, learning, trust, technical resources, and the difficulty of using AI to improve higher education. Considering the importance of trust and efficient knowledge management in handling uncertainty, the study emphasized the necessity of ongoing learning and technological resources for improving higher education capabilities.

Furthermore, Sysoyev [48] focused on assessing university faculty members' knowledge, preparedness, and use of AI technology in their professional practices. The study, which involved surveying 426 educators at 18 Russian universities, found that the educators are still mostly unaware of all the technology's potential. The conclusion made clear that, even though many university teachers claim to be neutral or open to integrating AI tools into their lessons, the application of these tools is restricted to technologies inside subject areas.

Similarly, Faraj [49] investigated the use of AI in higher education, focusing on Prince Sattam bin Abdulaziz University's development of future skills. The study employed a descriptive methodology, surveying 150 faculty members, and emphasized the significance of AI in education for sustainable development. In five domains—the learning environment, teachers, courses, students, and graduates—it defined essential requirements. The study's conclusion suggested the usage of AI to improve students' future talents.

The research gap identified is the limited investigation into the practical deployment and systematic integration of AI in higher education. While the potential and favorable perceptions of AI are recognized, there is a deficiency in conducting comprehensive inquiries into the practical efficacy and utilization of AI tools in diverse educational contexts. This study is crucial as it demonstrates the crucial role of self-professionalism in today's generative AI-driven world, emphasizing the need for ELT practitioners to stay current with new developments in English pedagogy. To underscore the research gap, particularly regarding the empowerment of EFL teachers' perceptions of generative AI-mediated self-professionalism, several scholars argue that Saudi initiatives for professional development within the EFL domain do not align with the optimal model of professional development [50, 51, 52, 53, 54, 55].

Consequently, English teachers "need to keep up with the information and skills necessary for their professional progress and be lifelong learners" [[56], p. 118]. The role of a language teacher has changed due to the adoption of new methods and techniques [57, 58, 59], affecting teaching practices, knowledge, understanding, self-awareness, beliefs, and attitudes [57,60]. Therefore, inspired by one of the highlights of Vision 2030 in Education, a comprehensive framework for the professional development of teachers and educational leaders, this research intends to bridge this gap by answering the following research questions:

1. To what extent are EFL teachers engaged in generative AI-mediated self-professionalism activities?

2. What are the EFL teachers' attitudes toward generative AI-mediated self-professionalism activities?

3. What are the constraints to engaging in generative AI-mediated self-professionalism activities?

4. What are the solutions to make the generative AI-mediated self-professionalism engagement effective?

## Methodology

### Research design

This study employed a mixed-methods research design to investigate EFL teachers' engagement in generative AI-mediated self-professionalism activities, their attitudes toward these activities, the constraints they face, and potential solutions to make the engagement effective. By utilizing quantitative and qualitative tools: a questionnaire and semi

structured interviews, the study ensured a well-rounded statistical and thematic analysis. The quantitative component assessed teachers' engagement levels and attitudes, facilitating statistical analysis of relationships between variables such as gender, years of experience, and types of professional development activities. This enabled the identification of trends, correlations, and general patterns across the sample. The qualitative component involved follow-up interviews with selected participants to explore the challenges faculty members encounter and identify solutions for enhancing generative AI-mediated self-professionalism. These insights enriched the quantitative findings by uncovering nuanced factors that quantitative analysis alone could not fully capture. This mixed-methods approach allowed triangulation, enhancing the validity and reliability of findings. By combining broad statistical patterns with in-depth contextual insights, the study provided a holistic exploration of EFL teachers' engagement, attitudes, challenges, and solutions related to generative AI in professional development.

## Participants

This study recruited EFL teachers from eight universities across Saudi higher education institutions, covering the northern, eastern, western, and southern regions. The recruitment period spanned from June 15, 2023, to August 15, 2023. The final sample consisted of 278 faculty members after excluding 22 incomplete or irrelevant responses from the initial 300 questionnaires distributed. The study sample included EFL faculty members of diverse nationalities, including but not limited to Australia, Algeria, Bangladesh, and other countries. Participants held various academic qualifications in English language studies, including master's and doctoral degrees in applied linguistics. Their teaching experience ranged from less than five years to over ten years. The faculty roles encompassed instructors, lecturers, assistant professors, associate professors, full professors, and administrative positions. Additionally, some participants possessed certifications in specialized areas within EFL education, such as language teaching methodologies and educational technology integration.

Moreover, a purposive sampling method was employed to ensure that participants met specific criteria related to the study objectives. The inclusion criteria for participation were as follows:

The participants should:

• be working as an EFL faculty member in a Saudi university

• hold a minimum of a master's degree in applied linguistics, TESOL, or a related discipline

• be engaged in EFL/EAP/ESP teaching and professional development activities

• hold teaching experience no less than one year

• have exposure to integrating technology in their instructional practices

Moreover, the exclusion criteria included non-EFL faculty, individuals without relevant academic qualifications and teaching experience, and incomplete survey responses.

Additionally, the recruitment of the participants included using official university channels, including institutional emails and faculty WhatsApp groups. The researchers circulated the questionnaire via email and WhatsApp groups, emphasizing the study's relevance to EFL professional development. Participation was voluntary, and no incentives were provided to avoid potential bias.

Furthermore, the study adhered to ethical guidelines to protect participants' rights and confidentiality. Approval was obtained from the Ethical Approval Committee at the Deanship Graduates Studies and Scientific Research, Najran University under the code [011157–024378-DS]. The key ethical measures included an informed consent, confidentiality, voluntary participation, and data security. For the informed consent, the questionnaire included an explicit consent text explaining the study's purpose, procedures, and potential risks. Participants were requested to acknowledge their informed consent before proceeding.

Also, to maintain confidentiality, the data was anonymized by assigning unique ID numbers to participants. Identifiable information, such as names and contact details, was securely stored in password-protected databases accessible only to the researchers. Additionally, all participants were informed of their right to withdraw at any stage without providing justification. Those involved in interviews signed two copies of the informed consent form, one retained by the researchers and the other by the participant. Finally, to ensure the security, the data was securely stored, and documentation of the informed consent process was maintained for future reference, ensuring transparency and adherence to ethical standards. The distribution of the study sample according to gender and years of experience is shown in Table 1.

### Instruments

**Data collection. Questionnaire:** The study employed a questionnaire to assess EFL teachers' engagement in generative AI-mediated self-professionalism activities and their attitudes toward such engagement. The questionnaire, developed by the researchers aligned with Mishra and Koehler's [61] Technological Pedagogical Content Knowledge (TPCK) framework, which integrates technology, pedagogy, and content in learning, in addition to insights from a literature review [14,56,62], consisted of three sections. The first section gathered personal information, including gender and years of teaching experience. The second section focused on generative AI-mediated self-professionalism activities, comprising 10 items, including EFL teachers' use of AI platforms for various professional development activities, participation in virtual coaching, and receiving guidance on teaching strategies, use AI tools to create engaging lesson plans, automate assessments, analyze student data, collaborate with peers, and maintain reflective practice journals. The third section explored EFL teachers' attitudes toward engagement in self-professionalism activities, comprising 20 items, including AI's help in strengthening EFL teachers' instructional efficacy, employing innovative teaching methods, integrating ICT, preparing lessons tailored to students' needs, creating student-friendly projects, providing feedback, building rapport and motivating learners, setting high learning targets, fostering critical thinking, developing authentic materials, addressing special needs, and handling classroom challenges.

The review of literature was utilized for the adaptation of various items of the questionnaire. For instance, "set my own learning goals to improve myself professionally" and "reflect on my practice as a teacher" from Asmari [56] were revised to "use AI platforms to assess my language teaching strengths, weaknesses, and professional goals." Similarly, Assia's [14] "self-assessment practices" and "using feedback from CPD" were modified to "track my progress and receive AI-generated feedback on my professional learning activities." Additionally, Bouaissane's [62] "peer collaboration" and "mentorship relationships" were modified to "participate in virtual coaching sessions facilitated by AI-powered chatbots or virtual mentors" under the theme of EFL teachers' engagement in generative AI-mediated self-professionalism. Furthermore, items such as "instructional practices" and "improving teaching efficacy" from Bouaissane [62] were adjusted to "helps in strengthening English language instructions and instructional efficacy." Also, Asmari's [56] "gaining new ideas to try out in the classroom" was modified as "assists in employing innovative ELT methods and strategies," and Assia's [14] "classroom management" was modified to "facilitates in using better management techniques in the classroom" under the theme of EFL teachers' attitudes toward generative AI-mediated self-professionalism engagement.

**Table 1. *Study sample's details.***

| Demographic variable | Group | No. | % |
|---|---|---|---|
| Gender | Male | 130 | 46.7 |
| | Female | 148 | 53.3 |
| Years of experience | 1-5 years | 80 | 28.7 |
| | 6-10 years | 100 | 36.1 |
| | + 10 years | 98 | 35.2 |
| Total | | 278 | 100 |

The decision to use a questionnaire aligns with Creswell's [63] assertion that "surveys help identify important beliefs and attitudes of individuals" (p. 06). A Google link to the questionnaire was made available for two weeks and distributed to participants through WhatsApp groups and emails. The average completion time for the questionnaire was approximately 15 minutes. To quantify responses, a five-point Likert scale was used: Always (5), Very often (4), Sometimes (3), Rarely (2), Never (1), and for teachers' attitudes: Strongly disagree (1), Disagree (2), Neutral (3), Agree (4), Strongly agree (5). This approach ensured consistency in responses and facilitated data analysis to gain insights into EFL teachers' perceptions and behaviors regarding generative AI-mediated self-professionalism activities.

Furthermore, to establish construct validity, the questionnaire underwent Exploratory Factor Analysis (EFA). EFA is a statistical method used in research to identify underlying relationships between observed variables and uncover latent constructs explaining correlation patterns [64]. EFA was conducted on the 210 participants' responses using Principal Component Analysis (PCA), one of the most accurate factor analysis techniques, as it extracts the maximum possible variance for each factor. The factor axes were orthogonally rotated using the Varimax with Kaiser Normalization method [65]. Items with factor loadings 0.3 or higher were considered to enhance scale validity [66].

Moreover, the first factor: EFL teachers' engagement in generative AI-mediated self-professionalism was a pure factor that accounted for 83.1% of the variance, while the second factor was also a pure factor that accounted for 55.7% of the variance, as illustrated in Fig 1.

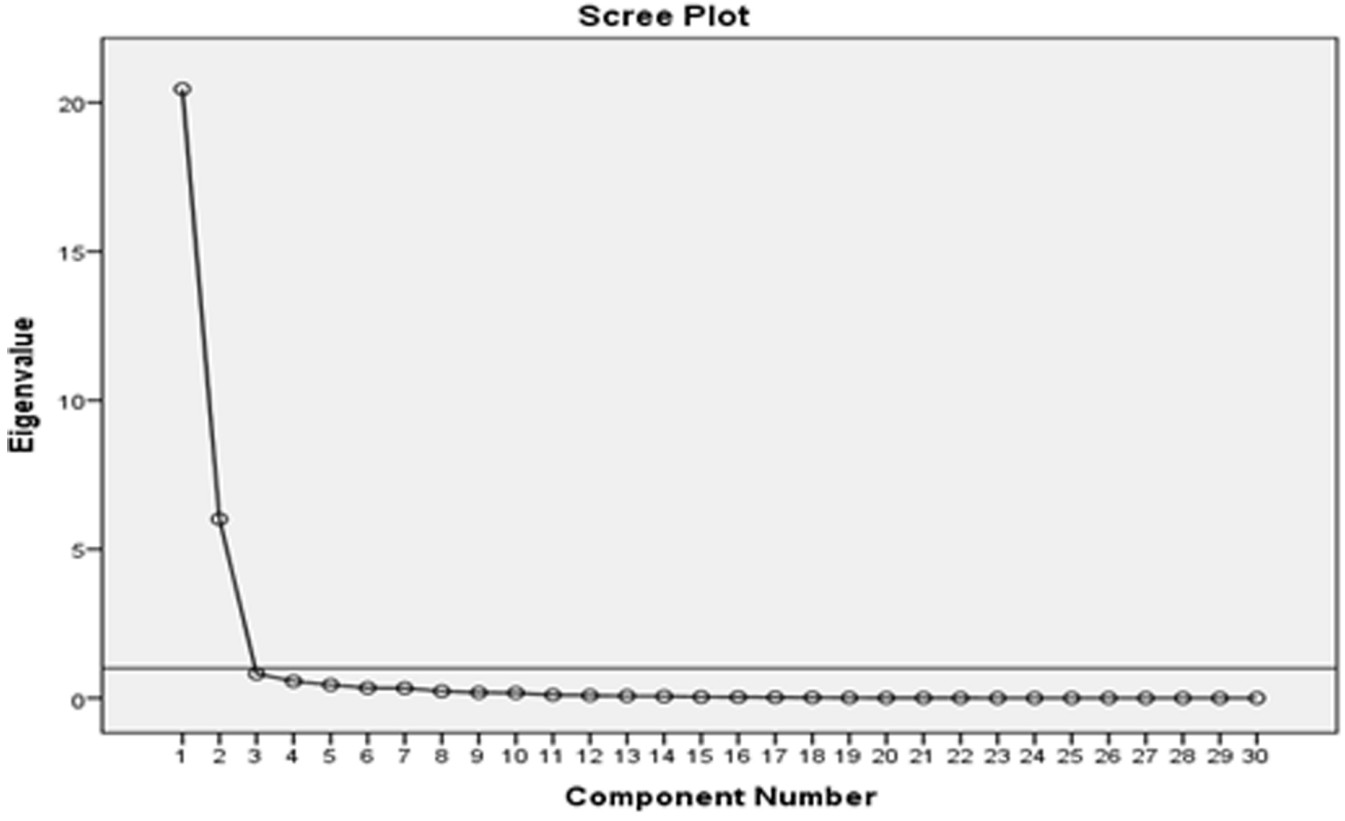

**Fig 1. Scree plot and the kaiser criterion (Eigenvalue >1), indicating a two-factor presentation.**

Table 2 presents the factor loadings suggest that both items contribute to the factor structure but with different weightings.

Fig 1 and Table 2 indicate that factor 1 has a strong positive loading on Item 1 (0.831) and a moderate negative loading on Item 2 (−0.557). This indicates that Factor 1 is closely related to Item 1 but inversely related to Item 2. Factor 2, on the other hand, has a strong positive loading on Item 2 (0.831) and a moderate positive loading on Item 1 (0.557). This suggests that Factor 2 is primarily associated with Item 2 but still has some connection with Item 1. Overall, Factor 1 and Factor 2 appear to represent contrasting yet related constructs. Furthermore, the factor structure suggests an implied relationship, meaning that while the two factors are distinct, they are not entirely independent.

Additionally, the Varimax with Kaiser Normalization rotation method was applied, and Table 3 shows the item loadings on the two extracted factors after rotation.

Table 3 illustrates that all scale items loaded onto the two factors, indicating that all scale items aligned into only two factors, which suggests the presence of an underlying theoretical construct. This serves as a valid indicator of the scale's construct validity.

Factor 1, a pure factor explaining 83.1% of the variance, included items a1 to a10.

Factor 2, a pure factor explaining 55.7% of the variance, included items b1 to b20.

## Semi-structured interview

In addition, a semi-structured interview was conducted to identify the constraints associated with engaging in generative AI-mediated self-professionalism activities, as well as the solutions to enhance the effectiveness of such engagements. Fifteen participants were chosen based on their willingness to share their experiences and the use of generative AI in their instructions. They were informed about the study's goals, procedures, and implications and were given informed consent. The interviews were conducted by the researchers using Blackboard, a digital communication platform, and audio recordings were made with participants' permission. Time management was well-maintained during the interviews, with each one estimated to last between six and eight minutes. The researchers used probing questions to probe further into participant responses.

Interviews were conducted between June 15, 2023, and August 15, 2023, with participants having at least one year of experience using generative AI. Interview questions were modified by a panel of experts to ensure quality and relevance, with probing questions used strategically to provide depth. Interviewers followed structured guidelines to maintain consistency across interviews. The interview duration was approximately six to eight minutes, which generally allowed for thorough coverage of all questions, though in some cases, time constraints may have limited the depth of responses. The interview questions included:

- What are the constraints to engaging in generative AI-mediated self-professionalism activities?
- What are solutions to make the generative AI-mediated self-professionalism engagement effective?

## Validity and reliability

This study examined the validity and reliability of the questionnaire and interview developed to assess EFL teachers' engagement with generative AI tools for self-professionalism. The contents of the study tools were assessed by the

**Table 2.** *Factor loadings after rotation.*

| Item | Factor 1: EFL teachers' engagement in generative AI-mediated self-professionalism | Factor 2: EFL teachers' attitudes toward generative AI-mediated self professionalism engagement |
|---|---|---|
| 1 | 0.831 | 0.557 |
| 2 | −0.557 | 0.831 |

**Table 3. *Rotated Component Matrix.***

| Item | Factors | |
| --- | --- | --- |
| | EFL teachers' attitudes toward generative AI-mediated self professionalism engagement | EFL teachers' engagement in generative AI-mediated self-professionalism |
| a1 | | .857 |
| a2 | | .960 |
| a3 | | .943 |
| a4 | | .931 |
| a5 | | .923 |
| a6 | | .960 |
| a7 | | .952 |
| a8 | | .940 |
| a9 | | .919 |
| a10 | | .930 |
| b1 | .970 | |
| b2 | .970 | |
| b3 | .970 | |
| b4 | .970 | |
| b5 | .970 | |
| b6 | .913 | |
| b7 | .884 | |
| b8 | .970 | |
| b9 | .970 | |
| b10 | .905 | |
| b11 | .949 | |
| b12 | .894 | |
| b13 | .694 | .447 |
| b14 | .577 | .424 |
| b15 | .701 | .459 |
| b16 | .903 | |
| b17 | .970 | |
| b18 | .635 | .528 |
| b19 | .758 | |
| b20 | .902 | |

Extraction Method: Principal Component Analysis.

Rotation Method: Varimax with Kaiser Normalization.

a. Rotation converged in 3 iterations.

experts to ensure face and content validity, covering aspects such as clarity, alignment with study objectives, relevance, and language appropriateness.

## Face validity

The content validity of the questionnaire and interview was assessed by a jury of five experts to ensure their validity in several aspects. The experts evaluated the compatibility of the statements with their respective domains, ensuring that the questions and interview topics aligned with the study's objectives. They also reviewed the appropriateness of the wording of the statements, ensuring clarity and relevance to the study's focus. Additionally, the experts assessed the inclusiveness

of the statements, verifying that they covered all relevant aspects of the topic. The experts also checked the language and grammatical soundness of the statements, ensuring they were clear and easily understood. Lastly, the experts considered the applicability of the statements in the context of EFL teachers' engagement with generative AI self-professionalism, confirming that the questions and topics were relevant and meaningful in this context.

The validity of the study tools was ensured by face validity and internal consistency. The experts were the faculty members who assessed the study tools and whether they could collect data to answer the study questions and thus achieve its objectives. The jury members, after multiple reviews, approved that the study tools could achieve the study objectives. Besides, some modifications related to wordiness, language, the study context, and items and domains were present. The following issues were observed by the experts:

| From | To |
|---|---|
| **EFL Teachers' engagement in generative AI-mediated self-professionalism** | |
| use AI platforms to assess my language teaching | use AI platforms to assess my language teaching strengths, weaknesses, and professional goals |
| track my progress and receive AI-generated feedback | track my progress and receive AI-generated feedback on my professional learning activities |
| participate in virtual coaching sessions | participate in virtual coaching sessions facilitated by AI-powered chatbots or virtual mentors |
| receive assistance and guidance | receive assistance and guidance from generative AI-mediated tools on my teaching strategies, classroom management techniques, lesson planning, and language assessment |
| receive personalized feedback and suggestions | receive personalized feedback and suggestions for improvement based on my teaching practices and student outcomes |
| integrate AI technologies | integrate AI technologies to automate the assessment and feedback process for student assignments and other assigned language learning tasks |
| utilize AI-driven analytics platforms | utilize AI-driven analytics platforms to analyze student data and identify patterns in language learning behavior and performance |
| participate in online communities | facilitated by AI platforms to collaborate with peers, share best practices, and provide feedback on teaching resources and strategies |
| – | use AI-powered platforms to maintain reflective practice journals, documenting my teaching experiences, challenges, and insights |
| **EFL teachers' attitudes toward generative AI-mediated self-professionalism** | |
| participate in online communities | facilitated by AI platforms to collaborate with peers, share best practices, and provide feedback on teaching resources and strategies |
| Helps in strengthening English instructions | Helps in strengthening English language instructions and instructional efficacy |
| Assists in ELT methods and strategies | Assists in employing innovative ELT methods and strategies |
| Facilitates in using better management techniques | Facilitates in using better management techniques in the classroom |
| Aids in understanding ICT use in EFL classrooms | Aids in understanding how to implement ICT in EFL classrooms |
| Assists in preparing lesson | Assists in preparing lesson to accommodate students' demands for language learning |
| Establish a cooperative language environment | Supports in establishing a cooperative language learning environment |
| Facilitates in creating student-friendly projects | Facilitates in creating student-friendly projects and assessments for language learning |
| Assists in adapting authentic ELT materials | Assists in adapting authentic ELT materials and resources |
| Helps in offering students useful feedback | Helps in offering students useful feedback and suggestions related to their language development |
| Assists in fostering strong rapport with students | Assists in fostering strong rapport with students through AI generated contents |
| Facilitates in motivating learners | Facilitates in motivating learners (both intrinsically and/or extrinsically) |
| Assists in setting learners' high-performance | Assists in setting learners' high-performance/language learning targets |
| Facilitates in activating students | Facilitates in activating students to maintain a lively classroom environment |
| Supports in fostering critical thinking | Supports in fostering critical thinking among students who take linguistics and literature courses |
| Helps in developing authentic materials | Helps in developing authentic materials and exercises with an emphasis on language development |
| Assists in identifying students with special needs | Assists in identifying students with special needs and development of tailored language learning plans |

| From | To |
|------|-----|
| Supports in language classroom challenges | Supports in handling language classroom challenges and settling issues |
| Facilitates in working together with others | Facilitates in working together with other peers (in virtual mentorship programs) and other stakeholders promote a positive language teaching and learning environment |
| Helps in knowing generative AI | Helps in knowing generative AI-mediated assistants to avail language teaching and learning opportunities |
| Facilitates in 21st century skills | Facilitates in fostering 21st century skills among language learners |

**Semi structured interview questions**
**From:**
-What are the issues in generative AI-mediated self-professionalism engagements?
-How to address those issues to make AI-mediated self-professionalism engagements effective?
**To:**
-What are the constraints to engaging in generative AI-mediated self-professionalism activities?
-What are solutions to make the generative AI-mediated self-professionalism engagement effective?

In addition, the study tool (questionnaire) was applied to a sample of (20) male and female participants for internal consistency. Pearson's correlation coefficient was then calculated between items, domain, and the whole scale. Table 2 presents the analysis results of the pilot study.

## Internal validity

Pearson's coefficients were applied to check the internal consistency of the questionnaire statements. The questionnaire was applied to a sample of (20) participants who were excluded from the main study later. The correlation was checked between statements and domains, and domains and the total degree of the questionnaire. Table 4 shows the values of Pearson's correlation coefficients.

Table 4 shows that Pearson correlation coefficients between the statements with their domain were statistically significant at the significance level (0.01). Pearson correlation coefficients ranged between the statements and the total score of the first domain (EFL teachers' generative AI engagement) between (0.619** − 0.875**). All values were significant at (0.01). Also, Pearson correlation coefficients between the statements (EFL teachers' attitudes towards generative AI) and the total score for the second domain ranged between (0.647** -- 0.887**) and were statistically significant at (0.01).

## Reliability

The reliability coefficients were calculated on the questionnaire domains using test-retest and Cronbach alpha methods. The Pearson correlation coefficients were calculated between the two applications. The study tool was applied to a survey sample of (25) teachers. Table 5 shows the reliability coefficients.

Table 5 shows that the reliability coefficient on the first domain reached (0.91) by Cronbach's alpha and (0.89) by test-retest. The second domain as well scored (0.93) using Cronbach's alpha and (0.90) using test-retest. These results indicate that the study tool is reliable.

## Data analysis

The statistical software SPSS, version 23, was utilized to analyze the results and address the research questions. To verify consistency, the Pearson correlation coefficient was computed. Additionally, Cronbach's alpha was employed to assess the reliability of the questionnaire. For answering the first two questions, measures such as means, standard deviations, and ranks were computed. A grading scale (Table 6) was applied to assess the degree of achievement of the statements and domains of the study tool.

**Table 4. Internal Consistency (Pearson).**

| No | Statement-domain | Pearson Correlation | Sig. | No | Statement-domain | Pearson Correlation | Sig. |
|---|---|---|---|---|---|---|---|
| | EFL teachers' engagement in generative AI-mediated self professionalism | | | | EFL teachers' attitudes toward generative AI-mediated self professionalism engagement | | |
| 1 | | .706** | .000 | 6 | | .647** | .000 |
| 2 | | .735** | .000 | 7 | | .658** | .000 |
| 3 | | .619** | .001 | 8 | | .648** | .000 |
| 4 | | .758** | .000 | 9 | | .753** | .000 |
| 5 | | .690** | .000 | 10 | | .753** | .000 |
| 6 | | .739** | .000 | 11 | | .807** | .000 |
| 7 | | .831** | .000 | 12 | | .864** | .000 |
| 8 | | .754** | .000 | 13 | | .815** | .000 |
| 9 | | .754** | .000 | 14 | | .828** | .000 |
| 10 | | .875** | .000 | 15 | | .877** | .000 |
| | EFL teachers' attitudes toward generative AI-mediated self professionalism engagement | | | 16 | | .882** | .000 |
| 1 | | .687** | .000 | 17 | | .850** | .000 |
| 2 | | .807** | .000 | 18 | | .828** | .000 |
| 3 | | .843** | .000 | 19 | | .887** | .000 |
| 4 | | .658** | .000 | 20 | | .845** | .000 |
| 5 | | .748** | .000 | | | | |

**Table 5. The study tool's reliability (questionnaire).**

| No. | Domain | Test-retest (Pearson) | Cronbach alpha |
|---|---|---|---|
| 1 | EFL teachers' engagement in generative AI-mediated self professionalism activities | 0.89 | 0.91 |
| 2 | EFL teachers' attitudes toward generative AI-mediated self professionalism engagement | 0.90 | 0.93 |

In addition, the qualitative data was analyzed following Braun and Clarke's [67] thematic analysis to detect patterns and structure results. Thematic analysis is widely recognized as a rigorous and systematic method for analyzing qualitative data, particularly in the context of interviews. This approach made it easier to find and compare responses, protecting anonymity and making data management more efficient. The process began with the transcriptions and the interviews were fully transcribed to preserve all verbal and non-verbal cues. Following the transcription, researchers familiarized themselves by repeatedly reading the transcripts, taking initial notes and highlighting key insights. In addition, the coding procedure was methodically implemented, assigning each participant a distinct code (e.g., T1, T2, T3) to ensure clarity, order, and convenience during the thematic analysis. The codes were grouped into broader themes reflecting key aspects of participants' experiences with generative AI. Furthermore, the segments of the interviews were thoroughly examined and sorted into sub-themes within each dimension, facilitating a thematic analysis of the data. Also, themes were reviewed, clearly defined, described, and illustrated with relevant examples to ensure they accurately represent the data. Moreover, the findings were contextualized in relation to the study objectives. Besides, data triangulation was utilized to improve the reliability and diligence of the results. Comparison between responses from the questionnaire and interview data was checked for consistency, showing both consistent and contrastive viewpoints. These methods enhanced the analysis by offering a more complete, multidimensional understanding of the respondents' perspectives, interactions, and behaviors regarding generative AI.

**Table 6.** *Degrees of approval.*

| Degree of approval | Very low | Low | Medium | High | Very high |
|---|---|---|---|---|---|
| Mean | 1-1.80 | >1.80-2.60 | >2.60-3.40 | >3.40-4.20 | >4.20-5.00 |

## Results

### RQ1: EFL Teachers' Engagement in Generative AI-Mediated Self-Professionalism Activities

Table 7 presents descriptive statistics of EFL teachers' engagement in generative AI-mediated self-professionalism activities. It shows that the participants were moderately engaged in these activities (M = 2.19, SD = .552). The data reveals varying levels of engagement among teachers, with some activities rated higher than others. For instance, using AI platforms to assess language teaching strengths, weaknesses, and professional goals was the most highly rated activity, indicating a high level of engagement. Activities such as maintaining reflective practice journals and participating in virtual coaching sessions also showed a moderate level of engagement. However, activities like integrating AI technologies for automated assessment and feedback processes and participating in online communities facilitated by AI platforms showed a lower level of engagement. Overall, the data suggests that while EFL teachers are engaged in some aspects of generative AI-mediated self-professionalism, there are areas where engagement could be improved.

### RQ2: EFL teachers' attitudes toward generative AI-mediated self-professionalism engagement

Table 8 presents descriptive statistics of EFL teachers' attitudes toward generative AI-mediated self-professionalism engagement. The data shows that the total degree of the study sample's responses to the EFL teachers' attitudes towards these activities was high (M = 4.12, SD = 1.035). This result indicates a generally positive attitude among teachers, with

**Table 7.** *Descriptive statistics of EFL teachers' engagement in generative AI-mediated self-professionalism.*

| No. | Statement<br>I, as an ELT practitioner, ……………. to strengthen my generative AI-mediated self-professionalism Engagement | Rank | Mean | Std. Deviation | Level |
|---|---|---|---|---|---|
| 1 | use AI platforms to assess my language teaching strengths, weaknesses, and professional goals | 1 | 4.03 | .742 | High |
| 2 | track my progress and receive AI-generated feedback on my professional learning activities | 5 | 3.06 | 1.339 | Medium |
| 3 | participate in virtual coaching sessions facilitated by AI-powered chatbots or virtual mentors | 3 | 3.31 | 1.001 | Medium |
| 4 | receive assistance and guidance from generative AI-mediated tools on my teaching strategies, classroom management techniques, lesson planning, and language assessment | 4 | 3.20 | 1.111 | Medium |
| 5 | receive personalized feedback and suggestions for improvement based on my teaching practices and student outcomes | 9 | 2.09 | 1.567 | Low |
| 6 | use AI-powered tools to create interactive and engaging lesson plans | 7 | 2.51 | 1.452 | Low |
| 7 | integrate AI technologies to automate the assessment and feedback process for student assignments and other assigned language learning tasks | 10 | 1.48 | 1.182 | Very low |
| 8 | utilize AI-driven analytics platforms to analyze student data and identify patterns in language learning behavior and performance | 6 | 2.90 | 1.118 | Medium |
| 9 | participate in online communities facilitated by AI platforms to collaborate with peers, share best practices, and provide feedback on teaching resources and strategies | 8 | 2.40 | 1.267 | Low |
| 10 | use AI-powered platforms to maintain reflective practice journals, documenting my teaching experiences, challenges, and insights | 2 | 3.74 | 1.236 | High |
| | Total degree | | 2.91 | .552 | Medium |

**Table 8.** *Descriptive statists of EFL teachers' attitudes toward generative AI-mediated self-professionalism engagement.*

| No. | Statement | Rank | Mean | Std. Deviation | Level |
|---|---|---|---|---|---|
| 1 | Helps in strengthening English language instructions and instructional efficacy | 3 | 4.34 | 1.200 | Very high |
| 2 | Assists in employing innovative ELT methods and strategies | 8 | 4.20 | 1.175 | Very high |
| 3 | Facilitates in using better management techniques in the classroom | 1 | 4.36 | 1.090 | Very high |
| 4 | Aids in understanding how to implement ICT in EFL classrooms | 2 | 4.35 | 1.093 | Very high |
| 5 | Assists in preparing lesson to accommodate students' demands for language learning | 4 | 4.33 | 1.201 | Very high |
| 6 | Supports in establishing a cooperative language learning environment | 17 | 3.91 | 1.164 | High |
| 7 | Facilitates in creating student-friendly projects and assessments for language learning | 15 | 3.97 | 1.239 | High |
| 8 | Assists in adapting authentic ELT materials and resources | 9 | 4.20 | 1.175 | Very high |
| 9 | Helps in offering students useful feedback and suggestions related to their language development | 5 | 4.31 | 1.198 | Very high |
| 10 | Assists in fostering strong rapport with students through AI generated contents | 19 | 3.86 | 1.120 | High |
| 11 | Facilitates in motivating learners (both intrinsically and/or extrinsically) | 18 | 3.91 | 1.349 | High |
| 12 | Assists in setting learners' high-performance/language learning targets | 20 | 3.81 | 1.207 | High |
| 13 | Facilitates in activating students to maintain a lively classroom environment | 11 | 4.16 | 1.211 | High |
| 14 | Supports in fostering critical thinking among students who take linguistics and literature courses | 14 | 3.97 | 1.021 | High |
| 15 | Helps in developing authentic materials and exercises with an emphasis on language development | 13 | 4.03 | 1.154 | High |
| 16 | Assists in identifying students with special needs and development of tailored language learning plans | 16 | 3.96 | 1.069 | High |
| 17 | Supports in handling language classroom challenges and settling issues | 6 | 4.26 | 1.188 | Very high |
| 18 | Facilitates in working together with other peers (in virtual mentorship programs) and other stakeholders promote a positive language teaching and learning environment | 12 | 4.09 | 1.236 | High |
| 19 | Helps in knowing GAI-mediated assistants to avail language teaching and learning opportunities | 7 | 4.20 | 1.071 | High |
| 20 | Facilitates in fostering 21st century skills among language learners | 10 | 4.19 | .997 | High |
| | Total | | 4.12 | 1.035 | High |

most statements rated as "Very high" or "High" in terms of agreement. Specifically, teachers believe that generative AI can help strengthen English language instruction, employ innovative ELT methods, facilitate better classroom management, and assist in understanding how to implement ICT in EFL classrooms. They also see generative AI as beneficial for preparing lessons, offering feedback to students, and fostering a cooperative learning environment. However, there are some areas where attitudes are slightly lower, such as in fostering 21st-century skills and using AI-generated content to build rapport with students. Overall, the data suggests that EFL teachers perceive generative AI as a valuable tool for enhancing their professional practices.

### RQ3: Constraints to engaging in generative AI-mediated self-professionalism activities

Teachers' responses to the constraints of engaging in generative AI-mediated self-professionalism activities were qualitatively analyzed. The findings showed that interviewees noted three major constraints: Technology Limitations, Knowledge and Skills, Ethical and Other Regulations.

### Technology limitations

The respondents were of the view that the implementation of AI for self-professionalism in EFL instructions faces challenges such as inadequate technology access, resource intensity, and sustainability. T1 added, "Not having the right

technology, like specific computers or programs, makes it hard to use AI for self-improvement." Infrastructure is crucial for effective AI use, while resource management is time-consuming. T12 said, "Using AI can take a lot of time and resources, which can be hard for teachers who are already busy." In addition, it may require additional financial investment. T14 told, "Making sure AI is working well is important but can be hard and might need more money."

### Knowledge and skills

The participants observed that EFL teachers face several barriers in using AI for self-professionalism, including lack of familiarity, skills gap, and data management issues. T3 added, "Not knowing how to use AI well and not understanding what it can do is a big problem for teachers trying to use AI for self-improvement." Furthermore, T4 said, "Not knowing much about AI and not having the skills to use it is a big challenge for teachers trying to use AI for self-improvement." These issues require digital literacy, targeted professional development, and technical skills to access and manage AI integration. T7 added, "AI needs a lot of information to learn, which can be tough for teachers trying to use AI for their own improvement."

### Ethical and other regulations

The interviewees' statements demonstrate that teachers face ethical, regulatory, and contextual issues when integrating AI for self-professionalism. T9 said, "Being ethical with AI is really important, especially when it comes to privacy and making sure the AI is fair." They need clear communication and training to navigate AI regulations. T10 added, "There are rules about using AI that teachers might not know about, which makes it hard for them to use AI." In addition, cultural contexts also pose constraints, making it very hard for EFL teachers to engage in generative AI-mediated self-professionalism activities. T15 said, "Different places and contexts have different ideas about AI, which can make it hard for teachers to use AI for self-improvement."

### RQ4: Solutions to make the generative AI-mediated self-professionalism engagement effective

Teachers' responses regarding the solutions to make generative AI-mediated self-professionalism effective were qualitatively analyzed. Participants suggested that generative AI-mediated self-professionalism can be improved by providing teachers with extensive education and training programs, making AI technology more accessible and affordable, and enhancing data accessibility through data-sharing agreements and cleaning tools. They were also of the view that adhering to clear ethical guidelines, staying informed about legal and regulatory requirements, collaborating with stakeholders, continuously evaluating and improving AI tools based on user feedback, and promoting diversity and inclusion in AI development and deployment can serve as solutions. Here are the selected excerpts under the sub-themes:

### Education and training

The participants identified two ways to improve the usability and accessibility of generative AI for EFL teachers. Teachers' knowledge and use of AI technologies will be enhanced by employing these strategies with an emphasis on targeted self-professionalism, which will also make AI technology more accessible and affordable. T1 added "One way to improve generative AI for teachers is to give them more education and training so they can understand and use AI better." Additionally, T2 added, "Making AI technology easier to use and more affordable would help. We should create simpler interfaces and offer AI services on the cloud to reduce the need for expensive hardware."

### Data sharing and ethical guidelines

The respondents emphasized the importance of data sharing, ethical guidelines, and institutional obligations in AI. They highlighted the need for collaborative data management and compliance with AI regulations. T6 added "Using agreements

for sharing data and tools to clean data can help improve the quality and accessibility of data for AI." The participants advocated for a comprehensive approach which can regulate ethical practices and legal issues to make the generative AI-mediated self-professionalism engagement effective. T8 told "Having clear ethical rules for using AI can reduce risks and ensure that AI is used responsibly and fairly." Moreover, T10 said "Institutions should know and follow the rules for using AI, consult legal experts, and protect data privacy and security."

### Collaboration and feedback mechanism

The participants emphasized the importance of collaborative, feedback-driven, and inclusive strategies in making the generative AI-mediated self-professionalism engagement impactful. T12 added "Working together with technology developers, educators, policymakers, and industry professionals can help develop and use AI better." They highlighted the need for collaboration among different stakeholders, gathering feedback from educators to create an effective, user-centric AI environment. T13 said "Getting feedback from teachers and using real-world data to improve AI tools is important for making them better." Furthermore, T15 told"Including diverse people in AI research teams and promoting diversity in AI development can help make AI better for everyone."

### Thematic framework

Tables 9 and 10 present a summary of the qualitative findings using thematic analysis framework, highlighting key themes, sub-themes, and supporting evidence (direct quotes).

## Discussion

The quantitative findings indicate that while ELT practitioners show interest in using AI for self-assessment and reflective practices, they are less engaged in AI-driven automation, student assessment, and interactive lesson planning. In addition, EFL teachers recognize AI as a valuable tool for improving instructional effectiveness, classroom management, and lesson planning, but slightly less so in areas related to student motivation, fostering collaboration, and supporting individual learning needs. The positive outcomes of generative AI-mediated self-professionalism, as per the findings of this study, show that they recognize the potential advantages of this approach, which is consistent with research by Nyaaba and Xiaoming [68], who found teachers are open to incorporating AI into their teaching methods. Mahdi's [30] study,

**Table 9. Constraints to engaging in AI-mediated self-professionalism.**

| Theme | Sub-Themes | Key Findings | Supporting Quotes |
|---|---|---|---|
| Technology Limitations | - Inadequate technology access – Resource intensity – Sustainability challenges | Teachers struggle due to a lack of required hardware, time constraints, and financial investment. | *"Not having the right technology, like specific computers or programs, makes it hard to use AI for self-improvement."* (T1) *"Using AI can take a lot of time and resources, which can be hard for teachers who are already busy."* (T12) *"Making sure AI is working well is important but can be hard and might need more money."* (T14) |
| Knowledge and Skills | - Lack of AI awareness – Skills gap – Data management issues | Many teachers lack AI literacy, making integration difficult. Professional training is needed. | *"Not knowing how to use AI well and not understanding what it can do is a big problem for teachers trying to use AI for self-improvement."* (T3) *"Not knowing much about AI and not having the skills to use it is a big challenge for teachers trying to use AI for self-improvement."* (T4) *"AI needs a lot of information to learn, which can be tough for teachers trying to use AI for their own improvement."* (T7) |
| Ethical and Other Regulations | - AI ethics & fairness – Legal & institutional barriers – Cultural constraints | Teachers are concerned about privacy, AI bias, and regulatory limitations that differ across contexts. | *"Being ethical with AI is really important, especially when it comes to privacy and making sure the AI is fair."* (T9) *"There are rules about using AI that teachers might not know about, which makes it hard for them to use AI."* (T10) *"Different places and contexts have different ideas about AI, which can make it hard for teachers to use AI for self-improvement."* (T15) |

**Table 10.** *Solutions for Effective AI-Mediated Self-Professionalism.*

| Theme | Sub-Themes | Key Findings | Supporting Quotes |
|---|---|---|---|
| Education and Training | - AI literacy programs – Affordable & user-friendly AI tools | Providing training and affordable AI technology can help teachers integrate AI into their self-professionalism efforts. | *"One way to improve generative AI for teachers is to give them more education and training so they can understand and use AI better."* (T1) *"Making AI technology easier to use and more affordable would help. We should create simpler interfaces and offer AI services on the cloud to reduce the need for expensive hardware."* (T2) |
| Data Sharing and Ethical Guidelines | - Institutional compliance – Ethical frameworks – Data-sharing policies | Establishing ethical AI usage guidelines and improving data access can enhance AI's effectiveness for professional development. | *"Using agreements for sharing data and tools to clean data can help improve the quality and accessibility of data for AI."* (T6) *"Having clear ethical rules for using AI can reduce risks and ensure that AI is used responsibly and fairly."* (T8) *"Institutions should know and follow the rules for using AI, consult legal experts, and protect data privacy and security."* (T10) |
| Collaboration and Feedback Mechanism | - Stakeholder collaboration – User feedback integration – Inclusive AI development | Working with policymakers, educators, and AI developers can lead to better AI tools and policies. | *"Working together with technology developers, educators, policymakers, and industry professionals can help develop and use AI better."* (T12) *"Getting feedback from teachers and using real-world data to improve AI tools is important for making them better."* (T13) *"Including diverse people in AI research teams and promoting diversity in AI development can help make AI better for everyone."* (T15) |

which emphasizes the value of individualized experiences in professional development, demonstrated a greater level of engagement in evaluating their teaching strengths and keeping reflective practice journals. Teachers' appreciation of AI's potential to improve reflective practices, according to the current study findings, aligns with the available literature [32], which recognizes the importance of CPD in promoting lifelong learning. Additionally, the findings of Al-Dosari's [46] study highlighted the gap in skill development and implementation methods for AI tools by the moderate participation in tracking progress and receiving feedback created by AI, as reflected in this study. The current study's findings in terms of teachers' reluctance to automate, although realizing AI's revolutionary potential, reflect larger worries in the cited literature regarding the quality of training, which frequently falls short of providing educators with the skills they need to successfully integrate AI [47].

The qualitative findings showed that teachers face challenges in generative AI-mediated self-professionalism due to limited access to technology, lack of AI proficiency, ethical concerns, legal requirements, time constraints, quality control issues, funding limitations, and varying cultural attitudes toward AI adoption. The current study findings accord with the previous research. For example, Baidoo-Anu and Owusu Ansah [69] showed hesitation to automate evaluation and feedback procedures which raises concerns about trust and understanding of AI technologies. This complexity points to a contradiction in which worries about AI's limitations and ethical issues impede its practical acceptance despite favorable sentiments. Furthermore, the study's findings in terms of the constraints, which include restricted access to technology, inadequate training, and ethical considerations, are consistent with the body of cited literature. Furthermore, the study's findings regarding AI adoption are consistent with Gill et al.'s [70] discussion, highlighting the many difficulties educators must overcome such as technology access, AI proficiency, ethical concerns, legal and time constraints, and faculty attitudes toward AI adoption.

Additionally, teachers emphasize the need for comprehensive training, accessibility, ethical guidelines, legal awareness, stakeholder collaboration, continuous AI improvement, and diversity in research to effectively integrate generative AI into their professional development and enhance teaching practices. The current finding is consistent with literature. For instance, Both Ng et al. [71] and Qadir [72] emphasize the necessity of AI competencies for effective integration into instructional strategies through training, ethical and legal awareness, collaboration, and diversity in research.

The following sections, research question wise, discuss how the results of the current study correspond with or differ from previous research while also highlighting the, triangulation, contributions of the current study, and theoretical and practical implications of the findings.

**RQ1: To what extent are EFL teachers engaged in generative AI-mediated self-professionalism activities?**

The results showed that EFL teachers show moderate engagement with generative AI for self-professionalism, with higher participation in self-assessment and reflective practices but lower engagement in automated assessment, feedback, and online collaboration. At the level of statements, the respondents highly used AI platforms to assess their language teaching strengths, weaknesses, and professional goals, including the use of AI-powered platforms to maintain reflective practice journals, documenting their teaching experiences, challenges, and insights. However, they only moderately tracked their progress and received AI-generated feedback on their professional activities. Additionally, they moderately participated in virtual coaching sessions facilitated by AI-powered chatbots or virtual mentors and received assistance and guidance from generative AI-mediated tools on their teaching strategies, classroom management techniques, lesson planning, and language assessment, including utilizing AI-driven analytics platforms to analyze student data and identify patterns in language learning behavior and performance. Sadly, they integrated AI technologies to automate the assessment and feedback process for student assignments and other assigned language learning tasks at a very low degree.

There may be several reasons why teachers' engagement in generative AI-mediated self-professionalism activities has not been satisfactory. Firstly, teachers may be less familiar with automating assessment and feedback processes for student assignments. Secondly, they may not perceive AI as useful in automating instructional processes. Thirdly, they may require additional training and support to fully integrate AI into their teaching practices. Finally, they may be less inclined to automate assessment and feedback processes due to limited benefits or concerns about drawbacks, including other constraints that hinder their adoption of AI technologies.

The result aligns with the current existing body of research that emphasizes the role of AI technologies in enhancing teachers' performance. Xu et al. [73] and Zheng et al. [74] reported positive outcomes of AI-powered language learning instructions on EFL learners' learning achievement. In addition, Herft [75] identified several ways that teachers can use AI applications like ChatGPT to support and improve their pedagogical and assessment practices. Furthermore, Faraj [49] suggested that the presence of prerequisites for implementing AI applications among faculty and students was only moderately common. This underscores the importance of more effective integration of AI to foster students' skills.

**RQ2: What are the EFL teachers' attitudes toward generative AI-mediated self-professionalism activities?**

The analysis revealed that the study participants held high attitudes toward generative AI-mediated self-professionalism activities, indicating a positive perception of the benefits of such engagement. Specifically, EFL teachers believed that these activities strengthen English language instruction and instructional efficacy, assist in employing innovative ELT methods and strategies, facilitate better classroom management, and aid in understanding how to implement ICT in EFL classrooms. They also perceived these activities as helpful in preparing lessons to meet students' language learning needs, establishing a cooperative learning environment, creating student-friendly projects and assessments, and adapting authentic ELT materials and resources.

Furthermore, teachers saw generative AI-mediated self-professionalism as beneficial for offering students useful feedback, fostering rapport, motivating learners, setting high-performance targets, maintaining a lively classroom environment, fostering critical thinking, developing authentic materials, identifying students with special needs, handling classroom challenges, and promoting a positive learning environment. They also recognized the role of AI in fostering 21st-century skills among language learners.

These findings suggest that EFL teachers perceive engagement in generative AI-mediated self-professionalism as advantageous, offering opportunities for professional growth, increased efficiency, and innovation in language teaching and learning. The result agrees relatively with the existing literature which highlights teachers' attitudes toward integrating technology in their profession. Nyaaba and Xiaoming [68] indicated that teacher educators in Ghana are willing to include GAI in their classroom practices. They believe that embracing GAI could revolutionize tasks like scoring, visualizing

lessons, research writing, and enhancing subject matter expertise. Furthermore, Kumar [76] underscored the significant advantages of AI grading, particularly in delivering timely feedback to students. Baidoo-Anu and Owusu Ansah [69] assert that ChatGPT offers various benefits, such as fostering personalized and interactive learning. Additionally, it generates prompts for formative assessment activities, thereby providing continuous feedback to enhance teaching and learning, among other advantages.

### RQ3: What are the constraints to engaging in generative AI-mediated self-professionalism activities?

The analysis showed that the teachers face constraints in generative AI-mediated self-professionalism activities in accessing the required technology, proficiency, and competency in using AI, and lack of generative AI literacy. The extensive data required for training and ethical considerations like data privacy and algorithmic bias are also significant challenges that the teachers view as constraints. Legal requirements and time-consuming integration of AI tools can also pose limitations. Efficient quality control procedures are essential but may require additional funds. Cultural attitudes towards AI and technology adoption also vary across regions, further complicating the process. The reasons for these findings may be attributed to the limited access to technology in addition to ethical considerations, legal requirements, time constraints, quality control, and limited funding. Cultural attitudes towards AI and technology adoption may be the reasons for these findings, making it difficult for some teachers to embrace AI.

The results of this study are consistent with the related research that focuses on constraints imposed by the use of AI in teaching practices. Ng et al. [71]suggest that teachers require competencies in AI to successfully integrate generative AI into classroom practices. Baidoo-Anu and Owusu Ansah [69] identified certain inherent limitations in ChatGPT. These limitations include the potential for generating incorrect information, biases in data training that may exacerbate existing biases, and privacy concerns. Wood et al. [77] identified gaps in understanding AI among participants. Qadir [72] highlights significant inherent limitations in the current state of ChatGPT. These limitations include the potential for generating incorrect answers and fabricating articles that do not exist. Gill et al. [70] and Samek and Müller [78] indicate that many teacher educators have a limited understanding of how generative AI operates. This lack of comprehension contributes to trust issues regarding its application. Zhai [79] asserts that assessment-related issues have been consistently present since the introduction of GAI tools like ChatGPT and GPT-4.

### RQ4: What are the solutions to make the generative AI-mediated self-professionalism engagement effective?

The analysis of teachers' responses regarding solutions to make generative AI-mediated self-professionalism more effective highlighted several key strategies. Participants emphasized the importance of providing teachers with extensive education and training programs to improve their proficiency in using generative AI for self-professionalism. They also emphasized the need to make AI technology more accessible and affordable, enhance data accessibility through data-sharing agreements and cleaning tools, and adhere to clear ethical guidelines. Additionally, they stressed the importance of staying informed about legal and regulatory requirements, collaborating with stakeholders, continuously evaluating and improving AI tools based on user feedback, and engaging diverse representation in research teams to promote diversity and inclusion in AI development and deployment. These findings suggest that teachers recognize the importance of comprehensive support and resources to effectively integrate generative AI into their professional development practices. By addressing these solutions, educators can enhance their use of generative AI for self-professionalism and ultimately improve their teaching practices and student outcomes. The results of this study align with previous research by Nyaaba and Xiaoming [68], which emphasized the critical knowledge teacher educators need for effective utilization of generative AI. Similarly, the findings coincide with those of Baidoo-Anu and Owusu Ansah [69], who highlighted the importance of collaboration among policymakers, researchers, educators, and technology experts to enhance education through the safe and effective use of evolving AI tools.

### Triangulation

The triangulation of quantitative and qualitative findings reveals both alignments and divergences in EFL teachers' engagement, attitudes, constraints, and proposed solutions regarding generative AI-mediated self-professionalism. The findings indicate that EFL teachers engage with AI tools for self-reflection and goal tracking (questionnaire), showing a preference for AI that supports introspective, personalized development (interview). Additionally, the analyses highlight positive attitudes toward AI's role in enhancing instructional efficacy and implementing innovative ELT methods (questionnaire). However, practical barriers such as limited resources, technological infrastructure, and ethical concerns consistently impact actual usage (interview), despite favorable attitudes.

However, deviations appear in engagement levels across tasks when the findings suggest high interest in using AI for varied tasks (questionnaire), while results emphasize practical gaps that hinder full engagement (interviews). Furthermore, low engagement in collaborative AI tasks (questionnaire) aligns with insights about time and resource constraints but contrasts with teachers' expressed interest in AI-driven collaboration if tools were more accessible (interview). The findings also highlight cultural and ethical concerns (interview), not fully captured in the questionnaire, suggesting that these may significantly influence engagement. Moreover, while quantitative analysis downplays solutions like training and collaboration, interview findings stress these as crucial for improving engagement, suggesting that more targeted support could bridge the divide between teachers' positive attitudes and actual AI engagement.

### Contribution of the study

This study makes several important contributions to the field of English language teaching (ELT) and teacher professional development, particularly in the context of generative AI adoption. First, the study provides empirical evidence on the extent to which EFL teachers engage with generative AI for self-professionalism, highlighting the areas where AI is most utilized (self-assessment and reflective practices) and where engagement remains low (automation, student assessment, and interactive lesson planning). Second, by exploring EFL teachers' perceptions, the study confirms a generally positive attitude toward generative AI in enhancing instructional effectiveness, classroom management, and lesson planning. However, it also identifies areas where AI is perceived as less effective, such as student motivation and fostering collaboration. Third, the research identifies key constraints that hinder AI integration in teachers' self-professionalism, including limited technological access, lack of AI proficiency, ethical and legal concerns, time constraints, and varying cultural attitudes toward AI adoption. These findings contribute to the ongoing discourse on AI adoption in education by providing a nuanced understanding of the barriers educators face. Fourth, the study proposes practical solutions for making AI-mediated self-professionalism more effective, including comprehensive teacher training programs, improved AI accessibility, clear ethical guidelines, legal awareness, and continuous evaluation of AI tools. These recommendations can inform policymakers, educational institutions, and AI developers in creating supportive frameworks for AI-enhanced teacher development. Finally, by investigating EFL teachers' engagement, attitudes, constraints, and solutions regarding generative AI-mediated self-professionalism, this study contributes to the limited body of research on AI's role in EFL teacher professional development, particularly in non-Western educational contexts.

### Implications

The findings of this study offer implications for educational institutions aiming to empower teachers' proficiency in utilizing generative AI tools through comprehensive training programs. These implications underscore the importance of prioritizing accessibility, affordability, and adherence to ethical considerations. Moreover, fostering collaboration among policymakers, researchers, educators, and technology experts is crucial. Continuous evaluation of AI tools and strategies to address teachers' challenges in engaging with AI-mediated self-professionalism activities is essential for effective implementation. Furthermore, the theoretical and practical implications derived from the result can be outlined in the following sections.

The research findings contribute to the theoretical implications of self-professionalism in the framework of learner agency and self-determined learning. The results suggest that while generative AI may help teachers advance their careers, its effectiveness will rely on how eager EFL teachers are to embrace and apply these technologies. This highlights the need for educational paradigms that facilitate the incorporation of technology into traditional frameworks for self-professionalism. Additionally, the study highlights the importance of sociocultural theories in understanding how context and culture impact instructors' usage of AI technologies. It suggests that addressing these cultural factors may be necessary to gain broad adoption of AI in self-professionalism.

From a practical implication standpoint, this study offers institutions and policymakers valuable information on how to enhance teachers' proficiency with generative AI technologies in EFL instructions. The focus on comprehensive training programs demonstrates the urgent need for structured professional development initiatives that are in line with instructors' real-world experiences and applications. The findings support a collaborative approach that includes stakeholders such as educators and IT experts to create an atmosphere that supports AI inclusion. This is particularly pertinent given the ethical concerns and other constraints, indicating that proactive measures must be taken to ensure that educators are equipped to manage these challenges. Furthermore, making AI technology accessible and affordable is also essential. Institutions should prioritize resources that enable teachers to engage with and experiment with AI tools. Examples of such projects include workshops, online resources, and continuous support systems that foster a culture of experimentation and feedback. Moreover, the implications for continuous evaluation of AI techniques and tools reinforce the need for an iterative approach to professional development. To enable changes based on feedback and experiences from the real world, educators must be involved in the ongoing development of AI mediated environments.

Accordingly, the researchers suggest that educational institutions can empower teachers by implementing comprehensive training programs, creating a framework for accessibility and affordability, and collaborating with researchers to conduct collaborative research studies. In addition, the institutions can design training programs and courses mediated by generative AI tools into the classroom, enhancing student engagement and personalizing learning experiences. Furthermore, EFL teachers can form professional learning communities to share best practices and foster a culture of self-professionalism. Moreover, institutions can also enhance self-professionalism initiatives by actively collecting feedback from EFL teachers and evaluating the effectiveness of AI tools in EFL classrooms to ensure that these tools remain relevant and impactful, addressing both teaching practices and the real-world challenges teachers face.

## Conclusion

The study aimed to enhance EFL teachers' perceptions of generative AI-mediated self-professionalism in Saudi higher education institutes. The results revealed moderate engagement in activities, such as self-assessment and virtual coaching sessions, but a high attitude towards AI-mediated self-professionalism. Activities like improving classroom management, integrating ICT into teaching, and creating customized lessons showed high engagement. However, constraints, such as limited access to technology, inadequate proficiency, and a lack of understanding of AI's capabilities were identified. Solutions to enhance AI-mediated self-professionalism include providing extensive education and training programs, making AI technology more accessible, improving data accessibility, adhering to ethical guidelines, collaborating with stakeholders, continuously refining AI tools, and promoting diversity and inclusion.

This study has several limitations that should be acknowledged. First, the generalizability of the findings is limited, as the study sample consists of EFL faculty members from only eight Saudi universities. While these institutions represent different regions of the country, the findings may not fully capture the perspectives of EFL educators in other higher education institutions within Saudi Arabia or in different cultural and institutional contexts. Additionally, the use of purposive sampling, while ensuring that participants met specific inclusion criteria, may have introduced selection bias. The study primarily included faculty members who actively engage in professional development and technology integration, potentially overlooking educators with limited exposure to AI-mediated self-professionalism. This limitation suggests that the

study may not fully represent the diversity of attitudes and experiences among all EFL faculty members in Saudi universities. Furthermore, the study exclusively focuses on EFL faculty members, excluding educators from other disciplines. While this focus allows for an in-depth exploration of AI in EFL self-professionalism, it does not provide insights into how faculty members in other fields perceive and utilize AI for their professional development. Moreover, the study is limited to higher education institutions, excluding EFL instructors in other educational settings such as language institutes, private schools, and vocational training centers. Investigating AI-mediated self-professionalism across different educational sectors could offer a more holistic perspective on the challenges and opportunities AI presents for language educators at various levels. Another limitation stems from the rapidly evolving nature of AI technologies. Given the continuous advancements in AI tools and their applications in education, the findings of this study may become outdated as new technologies emerge and educational policies evolve. Finally, cultural and institutional differences among participants could have influenced their engagement levels and attitudes toward AI. While the study includes faculty members from diverse national backgrounds, variations in university policies, access to technology, and institutional support for AI integration may have shaped their responses.

Future studies could address several gaps identified in this research. Expanding the study to include EFL faculty members from a broader range of institutions across different countries and educational contexts would provide a more comprehensive understanding of AI's role in faculty development. Additionally, representative sampling methods should be used to ensure more diverse and generalizable results. Longitudinal research would help track the long-term impact of AI tools on professional development over time. Moreover, further research could compare different generative AI tools beyond ChatGPT to gain a more comprehensive understanding of their roles in supporting EFL teachers' self-professionalism. By examining various AI platforms, scholars can explore how each tool contributes to different aspects of professional enrichment. In addition, incorporating perspectives from students, teacher educators, and educational leaders would provide a more holistic understanding of AI-mediated self-professionalism. Comparative studies across different cultural and institutional settings would help identify contextual factors, such as variations in instructional approaches, technology infrastructure, and digital regulation.

## Author contributions

**Conceptualization:** Mohd Nazim, Ali Abbas Falah Alzubi.

**Data curation:** Ali Abbas Falah Alzubi.

**Formal analysis:** Ali Abbas Falah Alzubi.

**Funding acquisition:** Mohd Nazim.

**Investigation:** Ali Abbas Falah Alzubi.

**Methodology:** Ali Abbas Falah Alzubi.

**Project administration:** Mohd Nazim.

**Validation:** Ali Abbas Falah Alzubi.

**Writing – original draft:** Mohd Nazim, Ali Abbas Falah Alzubi.

**Writing – review & editing:** Mohd Nazim, Ali Abbas Falah Alzubi.

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
