## [Decision Letter · Decision Letter 0]

Dear Dr. Nazim,

Thank you for submitting your manuscript to PLOS ONE. After careful consideration, we feel that it has merit but does not fully meet PLOS ONE’s publication criteria as it currently stands. Therefore, we invite you to submit a revised version of the manuscript that addresses the points raised during the review process.

We look forward to receiving your revised manuscript.

Kind regards,

Ömer Gökhan Ulum

Academic Editor

PLOS ONE

3. We note that you have referenced (

Atikler, A. (1997). The role of action research in self-development of an ELT teacher: A descriptive case study [Unpublished master's thesis] Bilkent University, Ankara.

Karaaslan, A.D. (2003). Teachers' perceptions of self-initiated professional development: A case study on Bakent University English Language teachers [Unpublished master thesis]. Middle East Technical University.

) which has currently not yet been accepted for publication. Please remove this from your References and amend this to state in the body of your manuscript: (ie “Bewick et al. [Unpublished]”) as detailed online in our guide for authors

Reviewers' comments:

Reviewer's Responses to Questions

**Comments to the Author**

1. Is the manuscript technically sound, and do the data support the conclusions?

Reviewer #1: No

Reviewer #2: Partly

2. Has the statistical analysis been performed appropriately and rigorously?

Reviewer #1: No

Reviewer #2: Yes

3. Have the authors made all data underlying the findings in their manuscript fully available?

Reviewer #1: Yes

Reviewer #2: No

4. Is the manuscript presented in an intelligible fashion and written in standard English?

Reviewer #1: Yes

Reviewer #2: No

Reviewer #1: The study addressed a timely topic. However, there are major issues with it. Please see my comments below:

- There are many sentences within the text needing relevant citations. Please fix this problem throughout the manuscript.

- The gap the study is going to address need to be more highlighted in the introduction section. Moreover, the significance of the study should be discussed for different stakeholders in this section.

- If the authors use a researcher-made questionnaire to gain the data, how did they measured the construct validity of the questionnaire?

- The all steps to gather the data need to be reported in details. For example, it should be reported how the authors conducted the semi-structured interview.

- More information about the data analysis should be provided. Unfortunately, the procedures reported to analyze the qualitative data are vague. For example, it is not clear how the authors measured the reliability and validity of the obtained findings.

- The required information about the tables need to be added. Moreover, please make sure the tables are set in line with the guidelines of APA 7the edition.

- The qualitative results need to be presented in a common why in which the required analysis among the excerpts should be added to the text.

- The discussion part is not critical. In line with the literature, the authors should be justified the obtained findings in tune with the literature. Moreover, please discuss the theoretical and practical implications of the results in this part.

- The implications of the findings should be discussed by giving tangible examples how they can be used by different stakeholders.

- Given the limitations imposed on the study further suggestions for future study should be illuminated.

- The language of the manuscript should be improved by doing more comprehensive proofreading.

Reviewer #2: 1. In the introduction section, it is recommended to highlight the innovative aspects of this study to emphasize its unique contributions to the field.

2. In the second part of the literature review, when discussing the relationship between teachers and technology, many statements lack supporting literature or cite outdated sources. It is advised to incorporate recent studies, such as:

Shadiev, R., Reynolds, B. L., & Li, R. (2024). The Use of Digital Technology for Sustainable Teaching and Learning. Sustainability, 16(13), 5353. https://doi.org/10.3390/su16135353

Rezai, A., Soyoof, A., & Reynolds, B. L. (2024). Informal digital learning of English and EFL teachers’ job engagement: Exploring the mediating role of technological pedagogical content knowledge and digital competence. System, 122, 103276. https://doi.org/10.1016/j.system.2024.103276

Gao, Y., Wang, Q., & Wang, X. (2024). Exploring EFL University Teachers’ Beliefs in Integrating ChatGPT and other Large Language Models in Language Education: A Study in China. Asia Pacific Journal of Education, 44(1), 29–44. DOI:10.1080/02188791.2024.2305173

3. In the questionnaire section, it is recommended to detail which original questionnaires were utilized and specify any modifications made based on the requirements of this study.

4. The content related to semi-structured interviews should be further expanded to provide a more comprehensive understanding of the interview process and findings.

5. It is suggested to add a dedicated section on data collection to clearly outline the procedures and methodologies employed in gathering data.

6. In the data analysis section, it is recommended to merge single sentences into cohesive paragraphs to enhance the clarity and flow of the analysis process.

7. In the research findings section, the amount of interview data presented under each dimension is excessive. It is advised to consolidate these interviews or distribute them across more dimensions for better organization and readability.

8. The discussion should be organized around the research questions and include relevant subheadings to ensure logical coherence and clarity in the analysis.

9. The final paragraph of the conclusion discusses pedagogical implications, which is also addressed in the last paragraph of the discussion section. It is recommended to merge these sections to avoid redundancy and streamline the presentation of pedagogical insights.

10. It is advised to limit each paragraph to a manageable number of words to improve the overall readability and maintain the reader’s engagement throughout the article.

**Do you want your identity to be public for this peer review?** For information about this choice, including consent withdrawal, please see our Privacy Policy

Reviewer #1: No

Reviewer #2: No

---

## [Author Response · Author response to Decision Letter 1]

7 Nov 2024

Dear Dr. Ulum,

Thank you for the opportunity to revise and resubmit our manuscript, Empowering EFL Teachers’ Perceptions of Generative AI-Mediated Self-Professionalism (PONE-D-24-28944). We appreciate the detailed feedback provided by you and the reviewers, which has significantly strengthened our study.

In response to the comments and suggestions, we have made the following revisions:

1. Reviewer #1’s Comments:

o We have added relevant citations throughout the manuscript to address areas needing additional references.

o The gap our study addresses has been highlighted in the introduction, and we expanded on the significance of the study for different stakeholders (pages 3–7).

o For construct validity, we provided detailed information regarding the researcher-made questionnaire (pages 16, 19–22).

o We clarified the data collection steps, including the semi-structured interview process (page 16).

o Additional information has been included on the qualitative data analysis and the methods used to establish reliability and validity (pages 27–28).

o The tables have been adjusted to meet APA 7th edition guidelines, and additional details were added.

o We improved the qualitative results by presenting analysis and interpretation alongside the selected excerpts (pages 32–36).

o The discussion section has been critically expanded to align with the literature, and we have addressed both theoretical and practical implications (pages 36–45).

o We discussed the implications for different stakeholders with specific examples (page 45).

o Suggestions for future research have been elaborated (page 47).

o We performed comprehensive proofreading to improve clarity and coherence.

2. Reviewer #2’s Comments:

o We revised the introduction to emphasize the innovative aspects of our study (pages 3–7).

o Recent studies were incorporated into the literature review to support statements about the relationship between teachers and technology (pages 9–11).

o In the questionnaire section, we clarified the origins and modifications of the questionnaires used (pages 16–17).

o The interview process details and the analysis of findings were expanded for clarity (pages 18, 32–36).

o A dedicated section for data collection was added to outline the methodologies employed (page 16).

o The data analysis section was revised for improved clarity and flow (page 26).

o In the research findings, we revised and distributed interview excerpts under sub-themes for better readability (pages 32–36).

o The discussion section was restructured with subheadings to align with research questions and improve coherence (pages 36–45).

o We streamlined pedagogical implications by merging relevant sections to avoid redundancy (page 48).

o Paragraphs were revised to maintain readability and engagement.

We have also uploaded all required documents, including:

• A rebuttal letter addressing each reviewer’s point.

• A marked-up version of the revised manuscript with track changes.

• A clean version of the revised manuscript without tracked changes.

Thank you again for this opportunity. We look forward to your response and hope our revisions meet the standards of PLOS ONE.

Kind regards,

---

## [Decision Letter · Decision Letter 1]

Thank you for submitting your manuscript to PLOS ONE. After careful consideration, we feel that it has merit but does not fully meet PLOS ONE’s publication criteria as it currently stands. Therefore, we invite you to submit a revised version of the manuscript that addresses the points raised during the review process.

We look forward to receiving your revised manuscript.

Kind regards,

Ömer Gökhan Ulum

Academic Editor

PLOS ONE

Reviewers' comments:

Reviewer's Responses to Questions

**Comments to the Author**

Reviewer #1: All comments have been addressed

Reviewer #2: (No Response)

Reviewer #3: (No Response)

2. Is the manuscript technically sound, and do the data support the conclusions?

Reviewer #1: No

Reviewer #2: Yes

Reviewer #3: Yes

3. Has the statistical analysis been performed appropriately and rigorously?

Reviewer #1: No

Reviewer #2: Yes

Reviewer #3: I Don't Know

4. Have the authors made all data underlying the findings in their manuscript fully available?

Reviewer #1: No

Reviewer #2: Yes

Reviewer #3: No

5. Is the manuscript presented in an intelligible fashion and written in standard English?

Reviewer #1: No

Reviewer #2: Yes

Reviewer #3: Yes

**Reviewer #1:**  Thank you for allowing me to review your manuscript. After careful consideration, I regret to inform you that I cannot recommend this paper for publication in its current form. The study suffers from significant methodological flaws and lacks the rigor required for publication in this journal. Please see my comments below:

1. The research questions suggest the need for a mixed-methods approach, as they aim to explore both quantitative and qualitative dimensions of the phenomenon under investigation. However, the study design does not reflect this requirement. The authors have not adequately justified their choice of methodology, nor have they demonstrated how the chosen design aligns with the research objectives. Furthermore, the methodology section is cluttered with irrelevant and redundant information, which detracts from the clarity and coherence of the study. A well-structured methodology section should clearly outline the research design, justify its appropriateness, and avoid unnecessary details.

2. The participants section is disorganized and lacks critical information. It should focus on key elements such as the demographic characteristics of the participants, the sampling method (e.g., random sampling, purposive sampling), inclusion and exclusion criteria, and the process of obtaining informed consent. Instead, the current section is filled with extraneous details that do not contribute to understanding the study population. Additionally, the authors have not clarified how they recruited participants or ensured ethical considerations, such as confidentiality and voluntary participation. These omissions raise concerns about the transparency and reproducibility of the study.

3. The authors claim to have used a researcher-made questionnaire but fail to provide a theoretical foundation for its design. A well-constructed questionnaire should be grounded in established theories or frameworks, and its development process should be clearly documented. Moreover, the authors have not addressed the construct validity of the questionnaire, which is typically assessed through exploratory factor analysis (EFA) and confirmatory factor analysis (CFA). Without evidence of construct validity, the credibility of the quantitative findings is severely compromised. This is a major flaw that undermines the scientific rigor of the study.

4. The qualitative data analysis section is inadequately described. The authors have not specified which data analysis method (e.g., thematic analysis, grounded theory, content analysis) was used, nor have they provided a clear step-by-step account of the analysis process. Additionally, there is no discussion of how the reliability and validity of the qualitative findings were ensured. For example, the authors could have employed strategies such as triangulation, member checking, or inter-coder reliability to enhance the trustworthiness of their results. The lack of transparency in this section makes it difficult to assess the quality of the qualitative findings.

5. The presentation of the qualitative results is subpar and does not meet the standards expected in high-quality journals. The findings are not organized in a coherent manner, and there is insufficient integration of direct quotes or illustrative examples to support the authors' interpretations. I recommend that the authors consult established journals in their field to learn how to effectively present qualitative findings, including the use of thematic frameworks, tables, and narrative explanations.

6. The discussion section fails to adequately contextualize the findings within the existing literature. The authors have not sufficiently explained how their results align with or diverge from previous studies, nor have they addressed the theoretical and practical implications of their findings. A strong discussion should critically engage with the literature, highlight the contributions of the study, and acknowledge its limitations. The current discussion is superficial and does not provide a meaningful synthesis of the results.

**Reviewer #2: ** The article has significantly improved after revisions, with only a few areas requiring further modification. A minor revision is recommended. Specific suggestions are as follows:

1. In the abstract, it is recommended to include the data analysis methods to provide readers with a clearer understanding of the study's approach.

2. In the last paragraph of the introduction, it is suggested that the author state the overarching objective of the study first, and then naturally lead into the research questions.

3. In the “Population and Sample” section, provide details on the inclusion criteria for participants. Additionally, the current single paragraph is overly lengthy—split it into two paragraphs for better readability and clarity.

4. In the conclusion, the final paragraph is redundant as it repeats earlier statements. It is recommended to delete this paragraph to avoid repetition.

**Reviewer #3:**  The paper is very interesting, yet the following are to be considered:

1.Editing is recommended.

2.The word ‘discovering’ at the beginning of the third question is better changed to ‘finding out’.

3.Table 1 should immediately come after the reference on page 13.

4.Table 1 is messy and confusing. It is better to expand it so the details appear separated.

5.No need for the definite article ‘The’ in the titles of tables or figures.

6.Table 2 is not situated under the sentence that stated this point on p.21. On p. 22, there is another reference to Table 2.

7.I think the researchers mean ‘internal’ rather than ‘international’ in “Pearson’s coefficients were applied to check the international consistency of the”.

8.Table 2 shows that only 10 respondents to the first domain and 20 to the second domain while the researchers’ statement in the previous page does not make this point clear. Fix this issue, please.

9.Sections in the Results section should be numbered to align with the pre-set research questions.

10.The constrains and solutions in the Results sections need to appear in terms of recurrent themes.

11.The conclusion section should be divided into four sections as replies to the research questions.

12.Do not use the ampersand ‘&’ within the text as you have in “Gill et al. (2024) and Samek & Müller (2019) indicate that”. Use it only within the in-text citation.

13.There is some redundancy in some sections including the Conclusion section.

14.The list of references is to be reconsidered due to having some inconsistency in some references.

**Do you want your identity to be public for this peer review?** For information about this choice, including consent withdrawal, please see our Privacy Policy

Reviewer #1: No

Reviewer #2: No

Reviewer #3: **Yes: ** Nawal Fadhil Abbas

---

## [Author Response · Author response to Decision Letter 2]

27 Mar 2025

Dear Reviewers,

Thank you for the opportunity to revise and resubmit our manuscript, Empowering EFL Teachers’ Perceptions of Generative AI-Mediated Self-Professionalism (PONE-D-24-28944). We appreciate the detailed feedback provided by you, which has significantly strengthened our study.

In response to the comments and suggestions, we have made the following revisions:

Reviewer #1: Thank you for allowing me to review your manuscript. After careful consideration, I regret to inform you that I cannot recommend this paper for publication in its current form. The study suffers from significant methodological flaws and lacks the rigor required for publication in this journal. Please see my comments below:

1. The research questions suggest the need for a mixed-methods approach, as they aim to explore both quantitative and qualitative dimensions of the phenomenon under investigation. However, the study design does not reflect this requirement. The authors have not adequately justified their choice of methodology, nor have they demonstrated how the chosen design aligns with the research objectives. Furthermore, the methodology section is cluttered with irrelevant and redundant information, which detracts from the clarity and coherence of the study. A well-structured methodology section should clearly outline the research design, justify its appropriateness, and avoid unnecessary details.

Response

The methodology section has been revised. It aligns with the research objectives and appropriately justifies the chosen research design. It now provides a clear rationale for the methodology, demonstrating how it supports the research questions. Additionally, redundant and irrelevant details have been removed to enhance clarity and coherence.

2. The participants’ section is disorganized and lacks critical information. It should focus on key elements such as the demographic characteristics of the participants, the sampling method (e.g., random sampling, purposive sampling), inclusion and exclusion criteria, and the process of obtaining informed consent. Instead, the current section is filled with extraneous details that do not contribute to understanding the study population. Additionally, the authors have not clarified how they recruited participants or ensured ethical considerations, such as confidentiality and voluntary participation. These omissions raise concerns about the transparency and reproducibility of the study.

Response

The participants’ section has been revised to ensure clarity and completeness. The section now includes detailed information on the demographic characteristics of the participants, the sampling method employed, as well as the inclusion and exclusion criteria. Additionally, the recruitment process and the ethical considerations undertaken have been outlined, which include confidentiality measures and voluntary participation.

3. The authors claim to have used a researcher-made questionnaire but fail to provide a theoretical foundation for its design. A well-constructed questionnaire should be grounded in established theories or frameworks, and its development process should be clearly documented. Moreover, the authors have not addressed the construct validity of the questionnaire, which is typically assessed through exploratory factor analysis (EFA) and confirmatory factor analysis (CFA). Without evidence of construct validity, the credibility of the quantitative findings is severely compromised. This is a major flaw that undermines the scientific rigor of the study.

Response

Keeping the importance of grounding the questionnaire in established theories and ensuring its validity, a theoretical foundation (TPCK framework) has been added. Additionally, a detailed explanation of the development process has also been provided. Furthermore, regarding construct validity, exploratory factor analysis (EFA) and results have been added.

4. The qualitative data analysis section is inadequately described. The authors have not specified which data analysis method (e.g., thematic analysis, grounded theory, content analysis) was used, nor have they provided a clear step-by-step account of the analysis process. Additionally, there is no discussion of how the reliability and validity of the qualitative findings were ensured. For example, the authors could have employed strategies such as triangulation, member checking, or inter-coder reliability to enhance the trustworthiness of their results. The lack of transparency in this section makes it difficult to assess the quality of the qualitative findings.

Response

A description of the qualitative data analysis procedure has been provided, along with a step-by-step outline of the thematic analysis. Additionally, all strategies employed to ensure the reliability and validity of the qualitative findings have been added.

5. The presentation of the qualitative results is subpar and does not meet the standards expected in high-quality journals. The findings are not organized in a coherent manner, and there is insufficient integration of direct quotes or illustrative examples to support the authors' interpretations. I recommend that the authors consult established journals in their field to learn how to effectively present qualitative findings, including the use of thematic frameworks, tables, and narrative explanations.

Response

The presentation of the qualitative results has been revised to enhance clarity, and coherence. The findings are now structured using a thematic framework, with improved organization and integration of direct quotes and illustrative examples to support the interpretations. Additionally, tables and narrative explanations have been incorporated

6. The discussion section fails to adequately contextualize the findings within the existing literature. The authors have not sufficiently explained how their results align with or diverge from previous studies, nor have they addressed the theoretical and practical implications of their findings. A strong discussion should critically engage with the literature, highlight the contributions of the study, and acknowledge its limitations. The current discussion is superficial and does not provide a meaningful synthesis of the results.

Response

The discussion section has been revised to better contextualize the findings of the current study within the existing literature. The analysis has been expanded to compare the current study’s results with previous research, highlighting both similarities and differences. Additionally, the theoretical and practical implications of the findings have been elaborated to ensure a more comprehensive discussion. Moreover, a dedicated section on the study’s contributions has been added, while the limitations have been acknowledged and refined in the conclusion section.

Reviewer #2: The article has significantly improved after revisions, with only a few areas requiring further modification. A minor revision is recommended. Specific suggestions are as follows:

1. In the abstract, it is recommended to include the data analysis methods to provide readers with a clearer understanding of the study's approach.

Response

The abstract has been revised, and the data analysis methods have been incorporated

2. In the last paragraph of the introduction, it is suggested that the author state the overarching objective of the study first, and then naturally lead into the research questions.

Response

In the last paragraph of the introduction, the objectives of the study have been added.

3. In the “Population and Sample” section, provide details on the inclusion criteria for participants. Additionally, the current single paragraph is overly lengthy—split it into two paragraphs for better readability and clarity.

Response

In the “Population and Sample” section, the inclusion criteria for participants have been provided.

4. In the conclusion, the final paragraph is redundant as it repeats earlier statements. It is recommended to delete this paragraph to avoid repetition.

Response

In the conclusion section, the final paragraph has been deleted.

Reviewer #3: The paper is very interesting, yet the following are to be considered:

1. Editing is recommended.

Response

The manuscript has been edited.

2. The word ‘discovering’ at the beginning of the third question is better changed to ‘finding out’.

Response

The word ‘discovering’ at the beginning of the third question is changed to ‘finding out’.

3. Table 1 should immediately come after the reference on page 13.

Response

Table 1 has been placed right after its reference.

4. Table 1 is messy and confusing. It is better to expand it so the details appear separated.

Response

Table 1 has been expanded.

5. No need for the definite article ‘The’ in the titles of tables or figures.

Response

The definite article ‘The’ in the titles of tables has been deleted.

6. Table 2 is not situated under the sentence that stated this point on p.21. On p. 22, there is another reference to Table 2.

Response

Table 2 (now table 4) has been fixed.

7. I think the researchers mean ‘internal’ rather than ‘international’ in “Pearson’s coefficients were applied to check the international consistency of the”.

Response

The word ‘international’ has been replaced with ‘internal’.

8. Table 2 shows that only 10 respondents to the first domain and 20 to the second domain while the researchers’ statement in the previous page does not make this point clear. Fix this issue, please.

Response

Please note that the table 2 (now table 4) presents 10 items/statements under the first domain and 20 items/statements to the second domain and not the participants.

9. Sections in the Results section should be numbered to align with the pre-set research questions.

Response

The sections in the ‘Results’ have been numbered to align with the research questions.

10.The constrains and solutions in the Results sections need to appear in terms of recurrent themes.

Response

The constraints and solutions in the Results section are organized into separate sections, as they correspond to two distinct research questions.

11. The conclusion section should be divided into four sections as replies to the research questions.

Response

To the researchers’ mind, this comment pertains to the ‘Discussion’ section, which has already been structured into sections corresponding to the research questions.

12.Do not use the ampersand ‘&’ within the text as you have in “Gill et al. (2024) and Samek & Müller (2019) indicate that”. Use it only within the in-text citation.

Response

This issue has been fixed.

13.There is some redundancy in some sections including the Conclusion section.

Response

The manuscript has been revised to eliminate all instances of redundancy, ensuring clarity and conciseness. This includes revising the Conclusion section to enhance coherence and avoid repetition. All necessary adjustments have been made to improve the overall flow and readability of the manuscript.

14.The list of references is to be reconsidered due to having some inconsistency in some references.

Response

The references section has been updated.

Thank you once again for this opportunity. We look forward to your positive response and hope our revisions meet the standards of PLOS ONE.

Kind regards,

---

## [Decision Letter · Decision Letter 2]

Dear Dr. Nazim,

Thank you for submitting your manuscript to PLOS ONE. After careful consideration, we feel that it has merit but does not fully meet PLOS ONE’s publication criteria as it currently stands. Therefore, we invite you to submit a revised version of the manuscript that addresses the points raised during the review process.

We look forward to receiving your revised manuscript.

Kind regards,

Ömer Gökhan Ulum

Academic Editor

PLOS ONE

Journal Requirements:

Reviewers' comments:

Reviewer's Responses to Questions

**Comments to the Author**

Reviewer #2: (No Response)

Reviewer #3: All comments have been addressed

2. Is the manuscript technically sound, and do the data support the conclusions?

Reviewer #2: Yes

Reviewer #3: Yes

3. Has the statistical analysis been performed appropriately and rigorously?

Reviewer #2: Yes

Reviewer #3: I Don't Know

4. Have the authors made all data underlying the findings in their manuscript fully available?

Reviewer #2: (No Response)

Reviewer #3: Yes

5. Is the manuscript presented in an intelligible fashion and written in standard English?

Reviewer #2: Yes

Reviewer #3: Yes

Reviewer #2: This manuscript has improved significantly after one round of revisions, with enhanced clarity, methodology, and structure. However, certain areas still require further refinement. Specific suggestions are outlined below.

1. In the first two paragraphs of the introduction, it is recommended to include more references from the past five years.

2. In the introduction, the last three paragraphs should be merged and concisely focus on the research design, innovations, and significance of this study.

3. In the literature review section, when discussing the application of AI in university language education, it is recommended to include some recent studies, such as:

-Rezai, A., Soyoof, A., & Reynolds, B. L. (2024). Disclosing the correlation between using ChatGPT and well‐being in EFL learners: Considering the mediating role of emotion regulation. European Journal of Education, 59(4), e12752. DOI: https://doi.org/10.1111/ejed.12752

-Wang, X., & Reynolds, B. L. (2024). Beyond the Books: Exploring Factors Shaping Chinese English Learners’ Engagement with Large Language Models for Vocabulary Learning.Education Sciences, 14(5), 496. DOI: https://doi.org/10.3390/educsci14050496

4. In the "Research Design and Context" section, it is recommended to remove the statement about the significance of the study at the end of the paragraph. This section should focus solely on the research design and context, as the significance of the study has already been addressed at the end of the introduction.

5. In the methodology section, it is suggested to follow the logic of "research design - participants – instruments-data collection - data analysis."

6. The presentation of the conclusion section has some issues. It is recommended to structure it according to three dimensions: summarizing the research findings, identifying research limitations, and suggesting future research directions.

Reviewer #3: Dear author,

Thanks for addressing most of the comments left, yet some other issues need to fix:

1. Editing is still needed.

2. The four objectives should appear in the -ing form (you have three infinitive and one gerund).

3. Since the figure or table are identified by the number, there is no need for the definite article 'the' as in "The figure (1) and table (2)". The same mistake is repeated in other places. Fix all, please.

4.Still there is no consistency in writing the sources in the reference list. When there is an article from a journal, a colon is used after the volume and issue followed by the beginning and ending of the pages. Check all, please.

5. Some titles are not italicized in the reference list.

**Do you want your identity to be public for this peer review?** For information about this choice, including consent withdrawal, please see our Privacy Policy

Reviewer #2: No

Reviewer #3: **Yes: ** Nawal Fadhil Abbas

---

## [Author Response · Author response to Decision Letter 3]

29 Apr 2025

Reviewer #2: This manuscript has improved significantly after one round of revisions, with enhanced clarity, methodology, and structure. However, certain areas still require further refinement. Specific suggestions are outlined below.

1. In the first two paragraphs of the introduction, it is recommended to include more references from the past five years.

Response

More references from the past five years, in the first two paragraphs of the introduction, have been included. (Pages 3-4)

2. In the introduction, the last three paragraphs should be merged and concisely focus on the research design, innovations, and significance of this study.

Response

In the introduction, the last three paragraphs have been merged with a focus on the research design, innovations, and significance of this study. (pages 6-7)

3. In the literature review section, when discussing the application of AI in university language education, it is recommended to include some recent studies.

Response

In the literature review, while discussing the application of AI in university language education, more recent studies have been included. (page 10)

4. In the "Research Design and Context" section, it is recommended to remove the statement about the significance of the study at the end of the paragraph. This section should focus solely on the research design and context, as the significance of the study has already been addressed at the end of the introduction.

Response

In the "Research Design and Context" section, the statement about the significance of the study at the end of the paragraph has been removed. (page 13)

5. In the methodology section, it is suggested to follow the logic of "research design - participants – instruments - data collection - data analysis."

Response

In the methodology section, the logic of "research design i.e. participants – instruments - data collection - data analysis has been followed. (pages 13-28)

6. The presentation of the conclusion section has some issues. It is recommended to structure it according to three dimensions: summarizing the research findings, identifying research limitations, and suggesting future research directions.

Response

The presentation of the conclusion section has been structured according to three dimensions: summary of the research findings, research limitations, and suggestions for future research directions. (pages 52-54)

Reviewer #3: Dear author,

Thanks for addressing most of the comments left, yet some other issues need to fix:

1. Editing is still needed.

Response

The manuscript has been revised and edited as needed.

2. The four objectives should appear in the -ing form (you have three infinitive and one gerund).

Response

The four objectives have been revised and now they appear in the -ing form. (page 8)

3. Since the figure or table are identified by the number, there is no need for the definite article. 'the' as in "The figure (1) and table (2)". The same mistake is repeated in other places. Fix all, please.

Response

Figures and tables have been revised, and all occurrences of the definite article ‘the’ have been removed and fixed. (page 18-21)

4.Still there is no consistency in writing the sources in the reference list. When there is an article from a journal, a colon is used after the volume and issue followed by the beginning and ending of the pages. Check all, please.

Response

All references have been checked, and colons have been replaced with commas.

5. Some titles are not italicized in the reference list.

Response

All references have been checked, and the titles have been italicized.

---

## [Editor Report · Decision Letter 3]

Dear Dr. Nazim,

Thank you for submitting your manuscript to PLOS ONE. After careful consideration, we feel that it has merit but does not fully meet PLOS ONE’s publication criteria as it currently stands. Therefore, we invite you to submit a revised version of the manuscript that addresses the points raised during the review process.

Reviewer 1

This manuscript has improved significantly after one round of revisions, with enhanced clarity, methodology, and structure. However, certain areas still require further refinement. Specific suggestions are outlined below.

1. In the first two paragraphs of the introduction, it is recommended to include more references from the past five years.

2. In the introduction, the last three paragraphs should be merged and concisely focus on the research design, innovations, and significance of this study.

3. In the literature review section, when discussing the application of AI in university language education, it is recommended to include some recent studies, such as:

-Rezai, A., Soyoof, A., & Reynolds, B. L. (2024). Disclosing the correlation between using ChatGPT and well‐being in EFL learners: Considering the mediating role of emotion regulation. European Journal of Education, 59(4), e12752. DOI: https://doi.org/10.1111/ejed.12752

-Wang, X., & Reynolds, B. L. (2024). Beyond the Books: Exploring Factors Shaping Chinese English Learners’ Engagement with Large Language Models for Vocabulary Learning.Education Sciences, 14(5), 496. DOI: https://doi.org/10.3390/educsci14050496

4. In the "Research Design and Context" section, it is recommended to remove the statement about the significance of the study at the end of the paragraph. This section should focus solely on the research design and context, as the significance of the study has already been addressed at the end of the introduction.

5. In the methodology section, it is suggested to follow the logic of "research design - participants – instruments-data collection - data analysis."

6. The presentation of the conclusion section has some issues. It is recommended to structure it according to three dimensions: summarizing the research findings, identifying research limitations, and suggesting future research directions.

Reviewer 2

Dear author,

Thanks for addressing most of the comments left, yet some other issues need to fix:

1. Editing is still needed.

2. The four objectives should appear in the -ing form (you have three infinitive and one gerund).

3. Since the figure or table are identified by the number, there is no need for the definite article 'the' as in "The figure (1) and table (2)". The same mistake is repeated in other places. Fix all, please.

4.Still there is no consistency in writing the sources in the reference list. When there is an article from a journal, a colon is used after the volume and issue followed by the beginning and ending of the pages. Check all, please.

5. Some titles are not italicized in the reference list.

We look forward to receiving your revised manuscript.

Kind regards,

Ömer Gökhan Ulum

Academic Editor

PLOS ONE
---

## [Author Response · Author response to Decision Letter 4]

3 Jun 2025

Reviewer #1:

This manuscript has improved significantly after one round of revisions, with enhanced clarity, methodology, and structure. However, certain areas still require further refinement. Specific suggestions are outlined below.

1. In the first two paragraphs of the introduction, it is recommended to include more references from the past five years.

Response

More references from the past five years, in the first two paragraphs of the introduction, have been included. (Pages 3-4)

2. In the introduction, the last three paragraphs should be merged and concisely focus on the research design, innovations, and significance of this study.

Response

In the introduction, the last three paragraphs have been merged with a focus on the research design, innovations, and significance of this study. (pages 6-7)

3. In the literature review section, when discussing the application of AI in university language education, it is recommended to include some recent studies.

Response

In the literature review, while discussing the application of AI in university language education, more recent studies have been included. (page 10)

4. In the "Research Design and Context" section, it is recommended to remove the statement about the significance of the study at the end of the paragraph. This section should focus solely on the research design and context, as the significance of the study has already been addressed at the end of the introduction.

Response

In the "Research Design and Context" section, the statement about the significance of the study at the end of the paragraph has been removed. (page 13)

5. In the methodology section, it is suggested to follow the logic of "research design - participants – instruments - data collection - data analysis."

Response

In the methodology section, the logic of "research design i.e. participants – instruments - data collection - data analysis has been followed. (pages 13-28)

6. The presentation of the conclusion section has some issues. It is recommended to structure it according to three dimensions: summarizing the research findings, identifying research limitations, and suggesting future research directions.

Response

The presentation of the conclusion section has been structured according to three dimensions: summary of the research findings, research limitations, and suggestions for future research directions. (pages 52-54)

Reviewer #2:

Dear author,

Thanks for addressing most of the comments left, yet some other issues need to fix:

1. Editing is still needed.

Response

The manuscript has been revised and edited as needed.

2. The four objectives should appear in the -ing form (you have three infinitive and one gerund).

Response

The four objectives have been revised and now they appear in the -ing form. (page 8)

3. Since the figure or table are identified by the number, there is no need for the definite article. 'the' as in "The figure (1) and table (2)". The same mistake is repeated in other places. Fix all, please.

Response

Figures and tables have been revised, and all occurrences of the definite article ‘the’ have been removed and fixed. (page 18-21)

4.Still there is no consistency in writing the sources in the reference list. When there is an article from a journal, a colon is used after the volume and issue followed by the beginning and ending of the pages. Check all, please.

Response

All references have been checked, and colons have been replaced with commas.

5. Some titles are not italicized in the reference list.

Response

All references have been checked, and the titles have been italicized.

---

## [Editor Report · Decision Letter 4]

Empowering EFL Teachers’ Perceptions of Generative AI-Mediated Self-Professionalism

PONE-D-24-28944R4

Dear Dr. Nazim,

We’re pleased to inform you that your manuscript has been judged scientifically suitable for publication and will be formally accepted for publication once it meets all outstanding technical requirements.

Kind regards,

Ömer Gökhan Ulum

Academic Editor

PLOS ONE
---

## [Editor Report · Acceptance letter]

PONE-D-24-28944R4

PLOS ONE

Dear Dr. Nazim,

I'm pleased to inform you that your manuscript has been deemed suitable for publication in PLOS ONE. Congratulations! Your manuscript is now being handed over to our production team.

Kind regards,

on behalf of

Dr. Ömer Gökhan Ulum

Academic Editor

PLOS ONE